# Macromolecular fungal ice nuclei in *Fusarium*: Effects of physical and chemical processing

Anna T. Kunert [1], Mira L. Pöhlker [1], Kai Tang [1], Carola S. Krevert [1], Carsten Wieder [1], Kai R. Speth [1], Linda E. Hanson [2], Cindy E. Morris [3], David G. Schmale III [4], Ulrich Pöschl [1], and Janine Fröhlich-Nowoisky [1]

[1]Multiphase Chemistry Department, Max Planck Institute for Chemistry, D-55128 Mainz, Germany
[2]USDA-ARS and Department of Plant, Soil and Microbial Science, Michigan State University, East Lansing, MI 48824, USA
[3]Plant Pathology Research Unit, INRA, 84143 Montfavet, France
[4]School of Plant and Environmental Sciences, Virginia Tech, Blacksburg, VA 24061, USA

**Correspondence:** Janine Fröhlich-Nowoisky (j.frohlich@mpic.de)

**Abstract.** Some biological particles and macromolecules are particularly efficient ice nuclei (IN), triggering ice formation at temperatures close to $0\,°C$. The impact of biological particles on cloud glaciation and the formation of precipitation is still poorly understood and constitutes a large gap in the scientific understanding of the interactions and co-evolution of life and climate. Ice nucleation activity in fungi was first discovered in the cosmopolitan genus *Fusarium*, which is widespread in soil and plants, has been found in atmospheric aerosol and cloud water samples, and can be regarded as the best studied IN-active fungus. The frequency and distribution of ice nucleation activity within *Fusarium*, however, remains elusive. Here, we tested more than 100 strains from 65 different *Fusarium* species for ice nucleation activity. In total, $\sim11\,\%$ of all tested species included ice nucleation-active (IN-active) strains, and $\sim16\,\%$ of all tested strains showed ice nucleation activity above $-12\,°C$. Besides *Fusarium* species with known ice nucleation activity, *F. armeniacum*, *F. begoniae*, *F. concentricum*, and *F. langsethiae* were newly identified as IN-active. The cumulative number of IN per gram of mycelium for all tested *Fusarium* species was comparable to other biological IN like *Sarocladium implicatum*, *Mortierella alpina*, and Snomax®. Filtration experiments indicate that cell-free ice-nucleating macromolecules (INMs) from *Fusarium* are smaller than 100 kDa, and that molecular aggregates can be formed in solution. Long-term storage and freeze-thaw cycle experiments revealed that the fungal IN in aqueous solution remain active over several months and in the course of repeated freezing and thawing. Exposure to ozone and nitrogen dioxide at atmospherically relevant concentration levels did also not affect the ice nucleation activity. Heat treatments at $40\,°C$ to $98\,°C$, however, strongly reduced the observed IN concentrations, confirming earlier hypotheses that the INM in *Fusarium* largely consists of a proteinaceous compound. The frequency and the wide distribution of ice nucleation activity within the genus *Fusarium*, combined with the stability of the IN under atmospherically relevant conditions, suggest a larger implication of fungal IN on the Earth's water cycle and climate than previously assumed.

## 1 Introduction

Ice particles in the atmosphere are formed either by homogeneous nucleation at temperatures below -38 °C or by heterogeneous nucleation catalyzed by particles or macromolecules serving as ice nuclei (IN) at warmer temperatures (Pruppacher and Klett, 1997; reviewed in detail in Fröhlich-Nowoisky et al. (2016) and Knopf et al. (2018)). Biological particles in particular are expected to play an important role as IN in the temperature range from -15 °C to 0 °C, but the impact of biological particles on cloud glaciation and the formation of precipitation is still poorly understood (Coluzza et al., 2017). Several studies suggest a triggering effect of biological IN for cloud glaciation and formation of precipitation (Creamean et al., 2013; DeMott and Prenni, 2010; Failor et al., 2017; Hanlon et al., 2017; Joly et al., 2014; Petters and Wright, 2015; Pratt et al., 2009; Stopelli et al., 2015, 2017), and former studies have shown that biological particles are more efficient than mineral IN (DeMott and Prenni, 2010; Després et al., 2012; Hill et al., 2014; Hoose and Möhler, 2012; Huffman et al., 2013; Möhler et al., 2007; Morris et al., 2014; Murray et al., 2012; Pratt et al., 2009).

The best characterized biological IN are common plant-associated bacteria of the genera *Pseudomonas*, *Pantoea*, and *Xanthomonas* (Garnham et al., 2011; Govindarajan and Lindow, 1988; Graether and Jia, 2001; Green and Warren, 1985; Hill et al., 2014; Kim et al., 1987; Ling et al., 2018; Šantl-Temkiv et al., 2015; Schmid et al., 1997; Wolber et al., 1986), and recently, an ice nucleation-active (IN-active) *Lysinibacillus* was found (Failor et al., 2017). The first identified IN-active fungi were strains of the genus *Fusarium* (Hasegawa et al., 1994; Pouleur et al., 1992; Richard et al., 1996; Tsumuki et al., 1992). To date, a few more fungal genera with varying initial freezing temperatures such as *Isaria farinosa* ($\sim$ -4 °C), *Mortierella alpina* ($\sim$ -5 °C), *Puccinia* species (-4 °C to -8 °C), and *Sarocladium* (formerly named *Acremonium*) *implicatum* ($\sim$ -9 °C) have been identified as IN-active (Fröhlich-Nowoisky et al., 2015; Huffman et al., 2013; Morris et al., 2013; Richard et al., 1996).

The genus *Fusarium* is cosmopolitan and includes saprophytes and pathogens of plants and animals (Leslie and Summerell, 2006; Nelson et al., 1994). Although they are considered to be primarily soil-borne fungi, many species of *Fusarium* are airborne (Prussin et al., 2014; Schmale et al., 2012; Schmale and Ross, 2015), and they were found in atmospheric and cloud water samples (e.g., Amato et al., 2007; Fröhlich-Nowoisky et al., 2009; Fulton, 1966). Some species can cause wilts, blights, root rots, and cankers in agriculturally important crops worldwide (e.g., Schmale and Gordon, 2003; Wang and Jeffers, 2000). Other species can produce secondary metabolites known as mycotoxins that can cause a variety of acute and chronic health effects in humans and animals (e.g., Bush et al., 2004; Ichinoe et al., 1983).

While the factors for a positive selective pressure for ice nucleation activity in *Fusarium* and other fungi have not been directly identified, an ecological advantage of initiating ice formation is easily conceivable. Indeed, most IN-active bacteria and fungi are isolated from regions with seasonal temperatures below 0 °C (Diehl et al., 2002; Schnell and Vali, 1972). Ice nucleation activity at temperatures close to 0 °C could be beneficial for pathogens or might provide an ecological advantage for saprophytic *Fusarium* species by facilitating in the acquisition of nutrients liberated during cell rupture of the host (Lindow et al., 1982). Furthermore, IN on the surface of the mycelium could avoid physical damage of the fungus by protective extracellular freezing (Fröhlich-Nowoisky et al., 2015; Zachariassen and Kristiansen, 2000) or to bind moisture as ice in cold and dry seasons (Pouleur et al., 1992). With increasing temperatures, the retained water can be of advantage in early vegetative

periods and for bacterial movement on the mycelial water film known as fungal highway (Kohlmeier et al., 2005; Warmink
et al., 2011). Moreover, ice nucleation activity might be beneficial for airborne *Fusarium* and for their return to the Earth's
surface under advantageous conditions in a feedback cycle known as bioprecipitation (Després et al., 2012; Morris et al., 2013,
2014; Sands et al., 1982). In addition, once the IN are released into the environment, they can adsorb to clay and might also be
available in the atmosphere associated with soil dust particles (Conen et al., 2011; Fröhlich-Nowoisky et al., 2015, 2016; Hill
et al., 2016; O'Sullivan et al., 2014, 2015, 2016; Sing and Sing, 2010).

The sources, abundance, and identity of biological IN are not well characterized (Coluzza et al., 2017), and it has been
proposed that systematic surveys will likely increase the number of IN-active fungal species discovered (Fröhlich-Nowoisky
et al., 2015). *Fusarium* is the best-known IN-active fungus, but the frequency and distribution of ice nucleation activity within
*Fusarium* is not well known. In this study, more than 100 strains from 65 different *Fusarium* species were tested for ice
nucleation activity in three laboratories with different freezing methods. A high-throughput droplet freezing assay was used to
quantify the IN of selected *Fusarium* species, and filtration experiments were performed to estimate the size of the *Fusarium*
IN. Furthermore, the stability of *Fusarium* IN upon exposure to ozone and nitrogen dioxide, under high and low or quickly
changing temperatures, and after short- and long-term storage under various conditions was investigated.

## 2 Materials and methods

### 2.1 Origin and growth conditions of fungal cultures

Thirty *Fusarium* strains from USDA-ARS/Michigan State University (L. Hanson, East Lansing, MI, USA), 13 strains from the
Schmale Laboratory at Virginia Tech (D. Schmale, Blacksburg, VA, USA), and 69 strains from the Kansas State University
Teaching Collection (J. Leslie, Manhattan, KS, USA) were screened for ice nucleation activity (Table S1).

The strains from the USDA-ARS/Michigan State University were collected from crop tissue (sugar beet). All isolates were
from field-grown beets and were obtained by hyphal tip transfer. The strains from the Schmale Laboratory at Virginia Tech were
collected with unmanned aircraft systems (UASs or drones) equipped with remotely-operated sampling devices containing a
*Fusarium* selective medium (e.g., Lin et al., 2013, 2014). All of the Schmale Laboratory strains were collected 100 m above
ground level at the Kentland Farm in Blacksburg, Virginia, USA. Detailed information is not available for the sources of the
strains for the Kansas State University Teaching collection. However, some of these strains are holotype strains referenced in
Leslie and Summerell (2006).

The stains from the USDA-ARS/Michigan State University were cultivated on dextrose peptone yeast extract agar, con-
taining 10 g L$^{-1}$ dextrose (VWR, Radnor, PA, USA), 3 g L$^{-1}$ peptone (Difco Proteose Peptone No. 3, Becton, Dickinson and
Company, Franklin Lakes, NY, USA), and 0.3 g L$^{-1}$ yeast extract (Merck, Kenilworth, NJ, USA), filtered through a 0.2 µm pore
diameter filter (PES disposable filter units, Life Science Products, Frederick, CO, USA). After filtration, 12 g L$^{-1}$ agarose (Cer-
tified Molecular Biology Agarose, Bio-Rad, Hercules, CA, USA) was added, and the medium was sterilized by autoclaving at
121 °C for 20 min. The colonies were grown at 22 °C to 24 °C for 7 to 19 days. The strains from the Schmale Laboratory at
Virginia Tech and the Kansas State University Teaching Collection were maintained in cryogenic storage at -80 °C and were

grown on quarter-strength potato dextrose agar (Difco Laboratories, Detroit, USA) on 100 mm petri plates at ambient room temperature for four days prior to ice nucleation assays.

For quantitative analysis, exposure experiments, heat treatments, freeze-thaw cycles, as well as short- and long-term storage tests a selection of IN-active tested strains was grown on full-strength potato dextrose agar (VWR International GmbH, Darmstadt, Germany) first at room temperature for four to six days and then at 6 °C for about four weeks. For filtration experiments, the fungal cultures were grown at 6 °C for up to six months.

## 2.2 Preparation and treatments of aqueous extracts

For LED-based Ice Nucleation Detection Apparatus (LINDA) (Stopelli et al., 2014) experiments (see Sect. 2.3), 4 mL of sterile 0.9 % NaCl was added to each of eight petri plates, and the fungal cultures were scraped with the flat end of a sterile bamboo skewer. The resulting suspension of mycelium and spores was filtered through a 100 μm filter (Corning Life Sciences, Reims, France).

For Twin-plate Ice Nucleation Assay (TINA) (Kunert et al., 2018) experiments (see Sect. 2.3) the fungal mycelium was scraped off the agar plate and transferred into a 15 mL tube (Greiner Bio One, Kremsmünster, Austria). The fresh weight of the mycelium was determined gravimetrically. Pure water was prepared as described in Kunert et al. (2018). Aliquots of 10 mL pure water were added before vortexing three times at 2 700 rpm for 30 s (Vortex-Genie 2, Scientific Industries, Inc., Bohemia, NY, USA) and centrifugation at 4 500 g for 10 min (Heraeus Megafuge 40, Thermo Scientific, Braunschweig, Germany). For all experiments the aqueous extract was filtered successively through a 5 μm and a 0.1 μm PES syringe filter (Acrodisc®, Sigma-Aldrich, Taufkirchen, Germany), and the aqueous extract contained IN from spores and mycelial surfaces.

For filtration experiments, the 0.1 μm filtrate was further filtered successively through 300 000 MWCO and 100 000 MWCO PES ultrafiltration units (Vivaspin®, Satorius AG, Göttingen, Germany). After each filtration step, the IN concentration was determined using TINA.

For exposure experiments, aqueous extracts of *F. acuminatum* 3-68 and *F. avenaceum* 2-106 were exposed to high concentrations of $O_3$ and $NO_2$ as described in Liu et al. (2017). Briefly, a mixture of 1 ppm $O_3$ and 1 ppm $NO_2$ was bubbled through 1 mL aliquots of aqueous extract for 4 h, and the IN concentration was determined using TINA.

For heat treatment experiments, aliquots of aqueous extracts of *F. acuminatum* 3-68, *F. armeniacum* 20970, *F. avenaceum* 2-106, and *F. langsethiae* 19084 were incubated at 40 °C, 70 °C, and 98 °C, respectively, for one hour. The IN concentration was determined using TINA.

For freeze-thaw cycles, the ice nucleation activity of *F. acuminatum* 3-68 was determined shortly after preparation of the aqueous extract and after storage at 6 °C for 24 h using TINA. Then, the aqueous extract was stored at -20 °C for 24 h and thawed again. The ice nucleation activity was tested before storage at -20 °C for an additional 24 h. After thawing, the ice nucleation activity was determined again.

For long-term storage experiments, the aqueous extract of various *Fusarium* species was stored at 6 °C for about four months or at -20 °C for about eight months, and the ice nucleation activity was determined using TINA.

## 2.3 Ice nucleation assays

Two independent droplet freezing assays conducted in two laboratories were used to investigate the distribution of ice nucleation activity within *Fusarium* in an initial screening.

First, a thermal cycler (PTC200, MJ Research, Hercules, CA, USA) was used as described in Fröhlich-Nowoisky et al. (2015) to screen 30 *Fusarium* strains from seven species from USDA-ARS/Michigan State University in the temperature range from -2 °C to -9 °C. Mycelium was picked with sterile pipette tips (Eppendorf, Westbury, NY, USA) into 80 µL aliquots of 0.2 µm pore diameter filtered dextrose peptone yeast (DPY) broth in sterile 96-well polypropylene PCR plates (VWR International, LLC, Radnor, PA, USA). Up to seven droplets were measured for each sample, and the mean freezing temperature was calculated. Aliquots of uninoculated DPY broth were used as negative controls, which did not freeze in the investigated temperature interval.

Second, the LED-based Ice Nucleation Detection Apparatus (LINDA) was used as described by Stopelli et al. (2014) to screen 13 strains from the Schmale Laboratory at Virginia Tech and 69 strains from the Kansas State University Teaching Collection. Aliquots of 200 µL of each aqueous extract were transferred to three separate 500 µL tubes and placed on ice for 1 h prior to the LINDA experiments. LINDA was run from -1 °C to -20 °C, and images of the samples were recorded every six seconds. The mean freezing temperature for three droplets was calculated. Note, that the aqueous extracts were prepared in 0.9 % NaCl solution, which could reduce the freezing temperatures by 0.5 °C based on theoretical calculations. We cannot exclude, however, that the high concentration of IN compensates the effect of NaCl on the freezing temperature. This is supported by the investigations of Stopelli et al. (2014), who did not find a systematic suppression of freezing at this salt concentration in LINDA experiments. As a negative control, a 0.9 % NaCl solution was added to three uninoculated agar plates, and the freezing started below -14 °C. As positive control, aqueous suspensions of *Pseudomonas syringae* CC94 from the collection of INRA (Avignon, France) (Berge et al., 2014) (with a final $OD_{580}$ of 0.5 to 0.7, i.e. $\sim 10^9$ bacteria mL$^{-1}$) were used for each experiment. The bacteria were grown on King's medium B (King et al., 1954) at 22 °C to 25 °C for 48 h, and aqueous suspensions were equilibrated at 4 °C for 1 h to 4 h before LINDA experiments. The freezing temperatures of *Pseudomonas syringae* CC94 ranged from -3.46 °C to -4.58 °C.

Ice nuclei of selected *Fusarium* species, which were long known for ice nucleation activity (*F. acuminatum*, *F. avenaceum*) as well as all the newly identified species, were further analyzed in immersion freezing mode using the high-throughput Twin-plate Ice Nucleation Assay (TINA) (Kunert et al., 2018). The aqueous extracts were serially diluted 10-fold with pure water by a liquid handling station (epMotion ep5073, Eppendorf, Hamburg, Germany) to a dilution where droplets remained liquid in the investigated temperature interval. Of each dilution, 96 droplets (3 µL) were tested with a continuous cooling rate of 1 °C min$^{-1}$ from 0 °C to -20 °C. Pure water samples (0.1 µm filtrated) served as a negative control for each experiment. These did not freeze in the observed temperature interval. The temperature was measured with an accuracy of 0.2 K (Kunert et al., 2018). The obtained fraction of frozen droplets ($f_{ice}$) and the counting error were used to calculate the cumulative number of IN ($N_m$) with the associated error using the Vali formula and the Gaussian error propagation (Kunert et al., 2018; Vali, 1971). For each experiment, the cumulative number of IN was averaged over all dilutions. If the experiment was repeated, the cumulative

number of IN was averaged over all experiments, and the standard error was calculated. Three independent experiments with aqueous extract from three individual fungal culture plates of the same strain showed similar results with only slight variation. An example of results is presented for *F. armeniacum* 20970 (Fig. S1).

## 3 Results and discussion

### 3.1 IN-active *Fusarium* species

Although several IN-active *Fusarium* species are known, the frequency and distribution of ice nucleation activity within the fungal genus *Fusarium* is still not well studied (Hasegawa et al., 1994; Humphreys et al., 2001; Pouleur et al., 1992; Richard et al., 1996; Tsumuki and Konno, 1994). Two initial screenings in the temperature range from -1 °C to -20 °C were performed to better evaluate the frequency of ice nucleation activity within *Fusarium*. A strain was defined as IN-active, when it initiated ice formation above - 9 °C (thermal cylcer) and -12 °C (LINDA), respectively.

In total, ∼ 16 % (18/112) of the tested strains showed ice nucleation activity with mean freezing temperatures of -3.5 °C to -11.2 °C (Table 1) in the typical range known for *Fusarium* (-1 °C and -9 °C) (Hasegawa et al., 1994; Humphreys et al., 2001; Pouleur et al., 1992; Richard et al., 1996; Tsumuki et al., 1992; Tsumuki and Konno, 1994). Most formerly reported initial freezing temperatures were obtained with different *Fusarium* strains, growth conditions, and freezing assays, which might explain differences compared to our results. The high proportion of IN-active strains within *F. acuminatum* is consistent with previous reports (Pouleur et al., 1992; Tsumuki et al., 1995). Overall, ∼ 11 % (7/65) of the tested species included IN-active strains. In addition to strains from *Fusarium* species with known ice nucleation activity, four *Fusarium* species were newly identified as IN-active: *F. armeniacum*, *F. begoniae*, *F. concentricum*, and *F. langsethiae*. In further experiments, the ice nucleation activity of *F. begoniae* and *F. concentricum* could not be verified.

The newly identified IN-active species are cosmopolitan. *Fusarium armeniacum* is a toxigenic saprophyte (Burgess et al., 1993) causing seed and root rot on soybeans (Ellis et al., 2012). The geographical distribution has been reported as tropical and subtropical (Leslie and Summerell, 2006), but it was also found in Minnesota, USA (Kommedahl et al., 1979) and Australia (Burgess et al., 1993). *Fusarium begoniae* is a plant pathogen of Begonia found in Germany with a potential wider distribution (Nirenberg and O'Donnell, 1998). *Fusarium concentricum* is a plant pathogen, frequently found in Central America and isolated from bananas (Aoki et al., 2001; Leslie and Summerell, 2006), and *F. langsethiae* is a broadly distributed cereal pathogen (Torp and Nirenberg, 2004). Some strains of these newly identified IN-active species are known to produce mycotoxins, which can threaten the health of humans and animals (Fotso et al., 2012; Kokkonen et al., 2012; Wing et al., 1993a, b).

The results suggest that the ice nucleation activity within *Fusarium* is more widespread than previously known. Not all *Fusarium* species include IN-active strains and not all strains within one species show ice nucleation activity. Earlier studies including experiments suggested that *Fusarium* IN are proteins or at least contain a proteinaceous compound (Hasegawa et al., 1994; Pouleur et al., 1992; Tsumuki and Konno, 1994). Their production requires energy, and we might assume that this trait would not be expressed or maintained unless there was an ecological advantage. It is known that *Fusarium* can regulate the gene expression for IN production depending on environmental conditions such as nutrient availability (Richard et al.,

1996), and some *Fusarium* species reduce or lose their ice nucleation activity after several subcultures (Pummer et al., 2013; Tsumuki et al., 1995). Thus, we cannot exclude that all *Fusarium* strains have the ability to produce IN. From the phylogenetic distribution of ice nucleation activity across the genus *Fusarium*, we can speculate that ice nucleation activity is a very old trait, but either the gene expression requires a trigger, which is not yet identified, or the trait might be in the process of being lost. It is unlikely, however, that the age of the genetic determinants of fungal ice nucleation activity is older than that in bacteria, since fungi diverged well after the age that has been attributed to the bacterial IN gene (Morris et al., 2014), and the genetic determinants are not the same as those in bacteria.

## 3.2   Quantification and size determination of IN from selected *Fusarium* species

A selection of IN-active *Fusarium* species was further investigated by extensive droplet freezing assay analysis using TINA. All tested *Fusarium* strains initiated ice nucleation between -3 °C and -4 °C (Fig. 1). Differences in the freezing temperatures between the initial screening and the quantitative analysis can be due to different growth conditions and freezing assays. The cumulative number of IN ($N_m$) per gram of mycelium was in the range between $10^8$ g$^{-1}$ and $10^{13}$ g$^{-1}$. *Fusarium acuminatum* 3-68 showed the highest ice nucleation activity and *F. langsethiae* the lowest per gram of mycelium. The results are comparable to other IN-active microorganisms like *Sarocladium implicatum* ($10^8$ g$^{-1}$, Pummer et al., 2015, *Mortierella alpina* ($10^9$ g$^{-1}$, Fröhlich-Nowoisky et al., 2015; $10^{10}$ g$^{-1}$, Kunert et al., 2018), and the bacterial IN-active substance Snomax® containing *Pseudomonas syringae* ($10^{12}$ g$^{-1}$, Budke and Koop, 2015; Kunert et al., 2018).

The size of the *Fusarium* IN was investigated by filtration experiments. Filtration through a 5 μm and a 0.1 μm filter did not affect the ice nucleation activity (Fig. 2), revealing that *Fusarium* IN are smaller than 100 nm, cell-free, easily removed from the fungus, and stay active in solution. This is in agreement with other *Fusarium* studies (O'Sullivan et al., 2015; Pouleur et al., 1992; Tsumuki and Konno, 1994). Moreover, biological INMs smaller than 200 nm were also found in various organisms e.g., other fungi (Fröhlich-Nowoisky et al., 2015; Pummer et al., 2015), leaves, bark, and pollen from birch trees (*Betula* spp.) (Felgitsch et al., 2018; Pummer et al., 2012), leaf litter (Schnell and Vali, 1973), some microalgae (Tesson and Šantl-Temkiv, 2018), strains of *Lysinibacillus* (Failor et al., 2017), and biological particles in the sea surface microlayer (Irish et al., 2019; Wilson et al., 2015). Filtration through a 300 000 MWCO filter unit decreased the cumulative number of IN per gram of mycelium about 50 % to 75 % depending on the *Fusarium* species, but a tremendous number of IN ($10^{10}$ - $10^{13}$ g$^{-1}$) still passed through the filter. The initial freezing temperature was slightly shifted towards lower temperatures. Further filtration through a 100 000 MWCO filter unit reduced the IN number to $10^8$ - $10^{10}$ g$^{-1}$, which is less than 1 % of the initial IN concentration. Additionally, the initial freezing temperature was shifted about one degree towards lower temperatures.

As ice nucleation activity was found in all filtrates, the aqueous extract of *Fusarium* consists of a mixture of IN-active proteins with different sizes. We hypothesize that *Fusarium* IN are macromolecules (INMs) smaller than 100 kDa, which agglomerate to large protein complexes in solution. Some of these complexes fall apart upon filtration, so that the INMs can pass through the filter. The small shift in the initial freezing temperature suggests that these INMs reassemble again to aggregates after filtration, as larger IN nucleate at warmer temperatures (Govindarajan and Lindow, 1988; Pummer et al., 2015). Erickson (2009) determined the size of proteins based on theoretical calculations. As the interior of proteins is closely

packed with no substantial holes and almost no water molecules inside, proteins are rigid structures with approximately the same density ($\sim 1.37\,\mathrm{g\,cm^{-1}}$). Assuming the protein as a smooth spherical particle, the minimum diameter of the INM would be smaller than 6.1 nm. Our results are in accordance with Lagzian et al. (2014), who cloned and expressed a 49 kDa IN-active protein from *F. acuminatum*.

As *Fusarium* IN are cell-free and can easily be washed off the fungal surface, they can be released in high numbers into the environment. If they are not degraded by microorganisms before, the IN can adsorb to soil dust and be aerosolized attached to these particles (Conen et al., 2011; Fröhlich-Nowoisky et al., 2015; Hill et al., 2016; O'Sullivan et al., 2014, 2015, 2016; Sing and Sing, 2010). This is in good agreement with Pruppacher and Klett (1997), who found a positive correlation between IN number concentration and particles in the coarse mode. Other releasing processes cannot be excluded, however, it is unlikely

that the INMs are present in the atmosphere as individual aerosol particles. Individual proteins with a diameter of $\sim 6$ nm, which may enter the atmosphere, would be in the nucleation mode size range, where particles tend to uptake gaseous compounds and grow to Aitken mode particles, which themselves tend to coagulate to larger agglomerates (Seinfeld and Pandis, 1998).

### 3.3    Stability of *Fusarium* IN

In the atmosphere, IN can interact with other aerosol particles or gases. They can be exposed to chemically modifying agents

like ozone and nitrogen dioxide, and physical stressors like high and low or quickly changing temperatures. To investigate the stability of *Fusarium* IN, we performed exposure experiments, heat treatments, freeze-thaw cycles, and long-term storage tests.

The influence of chemical processing on the *Fusarium* IN, in particular oxidation and nitration reactions as occurring during atmospheric aging, was investigated by exposing aqueous extracts from *F. acuminatum* 3-68 and *F. avenaceum* 2-106 to high concentrations of ozone and nitrogen dioxide in liquid phase. Figure 3 shows that for both species neither the initial freezing

temperature nor the cumulative number of IN per gram of mycelium was affected by exposure. These results demonstrate a high stability of *Fusarium* IN under oxidizing and nitrating conditions. This is in contrast to other biological IN e.g., bacterial IN (Snomax®) (Kunert et al., 2018), birch and alder pollen (Gute and Abbatt, 2018), and dissolved organic matter (Borduas-Dedekind et al., 2019), where exposure to oxidizing agents reduced the IN activity.

The stability of the INM in *Fusarium* was investigated in heat treatment experiments. The ice nucleation activity was reduced

significantly at a 40 °C treatment (Fig. 4). Between 40 % and 90 % of IN were lost at this temperature depending on the species, which supports the hypothesis that the INM in *Fusarium* consists of a proteinaceous compound. A heat treatment at 70 °C reduced the ice nucleation activity to less than 0.01 % compared to the initial level. Moreover, the initial freezing temperature was shifted to lower temperatures indicating a breakdown of the large protein aggregates. After a 98 °C treatment, we still found ice nucleation activity for all investigated species except for *F. avenaceum* 2-106. The results are in agreement

with previous studies, which also reported a reduction in ice nucleation activity with increasing temperature in heat treatment experiments (Hasegawa et al., 1994; Pouleur et al., 1992; Tsumuki and Konno, 1994). The remaining activity after the 98 °C treatment, however, could indicate that post-translational modifications like glycosylation and therefore polysaccharides could play a role in the ice nucleation activity of *Fusarium*. Further systematic studies including chemical analyses are needed for elucidation.

To study the effects of short-term storage and freeze-thaw cycles on the ice nucleation activity of *F. acuminatum* 3-68, IN measurements of the same aqueous extract were performed at different time points (Fig. 5). The results of freshly prepared aqueous extract revealed that the highest activity of fungal IN was already developed during preparation of the filtrate and no time for equilibration was required. Storage of aqueous extract at 6 °C for 24 h did not affect the ice nucleation activity. Also, further storage at -20 °C for another 24 h, and repeated freeze-thaw cycles had no impact on the ice nucleation activity. This means, that, once in the atmosphere, the IN can undergo several freeze-thaw cycles without losing their activity and are still able to influence cloud glaciation and the formation of precipitation. This could be an explanation why not all fungi are always IN-active as their IN are highly stable and quasi recyclable. Ice nuclei might influence the availability of moisture over long times periods, and if enough moisture is available in the environment, the necessity of IN production would be omitted and the fungus could save energy.

In addition, the stability of the INM in *Fusarium* was studied in long-term storage tests, where aqueous extracts of various *Fusarium* species were stored at different temperatures for a long period of time. Figure 6 shows that storage at 6 °C for four months and -20 °C for eight months, respectively, did not influence the ice nucleation activity of *F. armeniacum* 20970, *F. acuminatum* 1-4, *F. avenaceum* 2-106, and *F. acuminatum* 2-38. The results demonstrate the high stability of the INMs in *Fusarium* in liquid and frozen solutions over long time periods, which makes *Fusarium* well applicable for laboratory IN studies. Moreover, the high stability is likely an advantage for these fungi to be linked to atmospheric processes.

## 4  Conclusions

The frequency and distribution of ice nucleation activity within the fungal genus *Fusarium* was investigated in a screening of more than 100 strains from 65 different *Fusarium* species. In total, $\sim 11\,\%$ (7/65) of all tested species included IN-active strains, and $\sim 16\,\%$ (18/112) of all tested strains showed ice nucleation activity, demonstrating the wide distribution of ice nucleation activity within *Fusarium*. Filtration experiments suggest that *Fusarium* IN form aggregates consisting of INMs smaller than 100 kDa ($\sim 6$ nm). Exposure experiments, freeze-thaw cycles, and long-term storage tests revealed a high stability of the INMs in *Fusarium*, demonstrating the suitability of *Fusarium* in laboratory IN studies. Heat treatments at 40 °C to 98 °C reduced the IN concentration significantly, supporting the hypothesis that the INM in *Fusarium* largely consists of a proteinaceous compound. An involvement of polysaccharides, however, cannot be excluded. The wide distribution of ice nucleation activity within the genus *Fusarium* together with the stability of the INM in *Fusarium* under atmospherically relevant conditions, suggest that the implication of these IN on the Earth's water cycle and climate might be more significant than previously assumed. Additional research is necessary to characterize the INMs in *Fusarium* and processes, which can result in their agglomeration to larger protein complexes. To evaluate the implication of these IN on the Earth's climate, additional work is required to study the abundance of *Fusarium* IN in environmental samples on a global scale.

*Data availability.*  All data are available from the corresponding authors upon request.

*Author contributions.* C.E.M., J.F.-N., U.P. designed the experiments. D.G.S. III and L.E.H. provided fungal cultures. C.E.M., D.G.S. III, and J.F.-N. performed the initial screenings. A.T.K., K.T., C.S.K., C.W., and K.R.S. performed the experiments. A.T.K., J.F.-N., M.L.P., and U.P. discussed the results. A.T.K. and J.F.-N. wrote the manuscript with contributions of all co-authors.

*Competing interests.* The authors declare that they have no conflict of interest.

*Acknowledgements.* We thank C. Bartoli, J.-D. Förster, T. Godwill, N.-M. Kropf, and E. Stopelli for technical assistance, G. D. Franc, T. C. J. Hill, K. Reinmuth-Selzle, B. Sánchez-Parra, J. F. Scheel, and M. G. Weller for helpful discussions, and the Max Planck Society (MPG), the Deutsche Forschungsgemeinschaft (DFG, FR3641/1-2, FOR 1525 INUIT) for financial support. This work is dedicated to the memory of Gary D. Franc, whose pioneering work in atmospheric microbiology has been an inspiration for this work.

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

**Table 1.** Ice nucleation-active *Fusarium* strains with corresponding mean freezing temperatures of the initial screening. The newly identified IN-active *Fusarium* species are marked with an asterisk (*).

| Species | Strain | $T$ (°C) |
|---|---|---|
| *F. acuminatum* | 1-3 | -5.6 |
| *F. acuminatum* | 1-4 | -5.0 |
| *F. acuminatum* | 1-5 | -5.6 |
| *F. acuminatum* | 1-24 | -3.5 |
| *F. acuminatum* | 2-38 | -5.0 |
| *F. acuminatum* | 2-48 | -5.6 |
| *F. acuminatum* | 2-109 | -5.6 |
| *F. acuminatum* | 3-48 | -5.0 |
| *F. acuminatum* | 3-68 | -3.5 |
| *F. acuminatum* | 20964 | -6.2 |
| *F. armeniacum** | 20970 | -5.3 |
| *F. avenaceum* | 2-106 | -5.0 |
| *F. avenaceum* | 11440 | -7.6 |
| *F. begoniae** | 10767 | -11.2 |
| *F. concentricum** | 10765 | -4.6 |
| *F. langsethiae** | 19084 | -9.4 |
| *F. tricinictum* | 20990 | -7.3 |

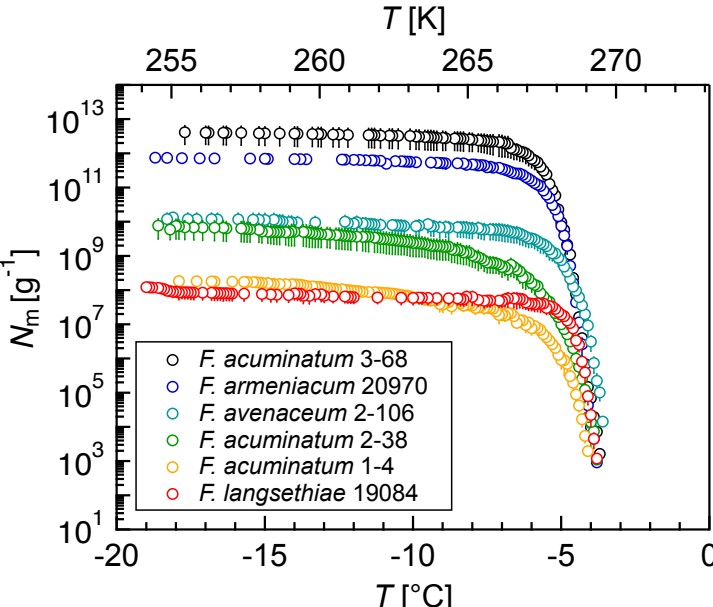

**Figure 1.** Overview of ice nucleation activity for selected *Fusarium* species and strains: cumulative number of IN ($N_m$) per gram of mycelium plotted against the temperature ($T$); arithmetic mean values and standard error of three independent experiments with aqueous extracts from different fungal culture plates.

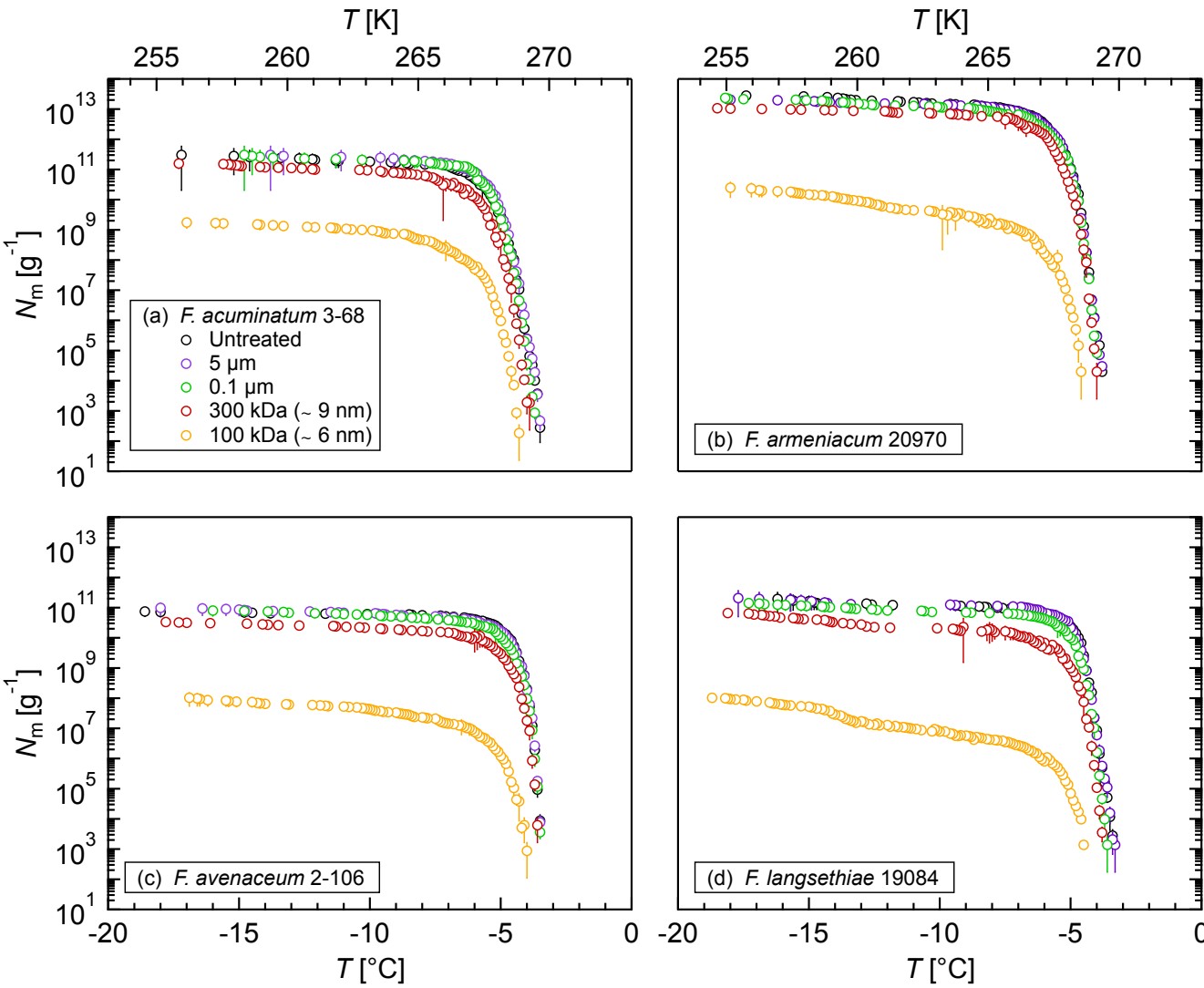

**Figure 2.** Size determination of the *Fusarium* IN upon filtration: cumulative number of IN ($N_m$) per gram of mycelium plotted against the temperature ($T$) for (a) *F. acuminatum* 3-68, (b) *F. armeniacum* 20970, (c) *F. avenaceum* 2-106, and (d) *F. langsethiae* 19084. The error bars were calculated using the counting error and the Gaussian error propagation.

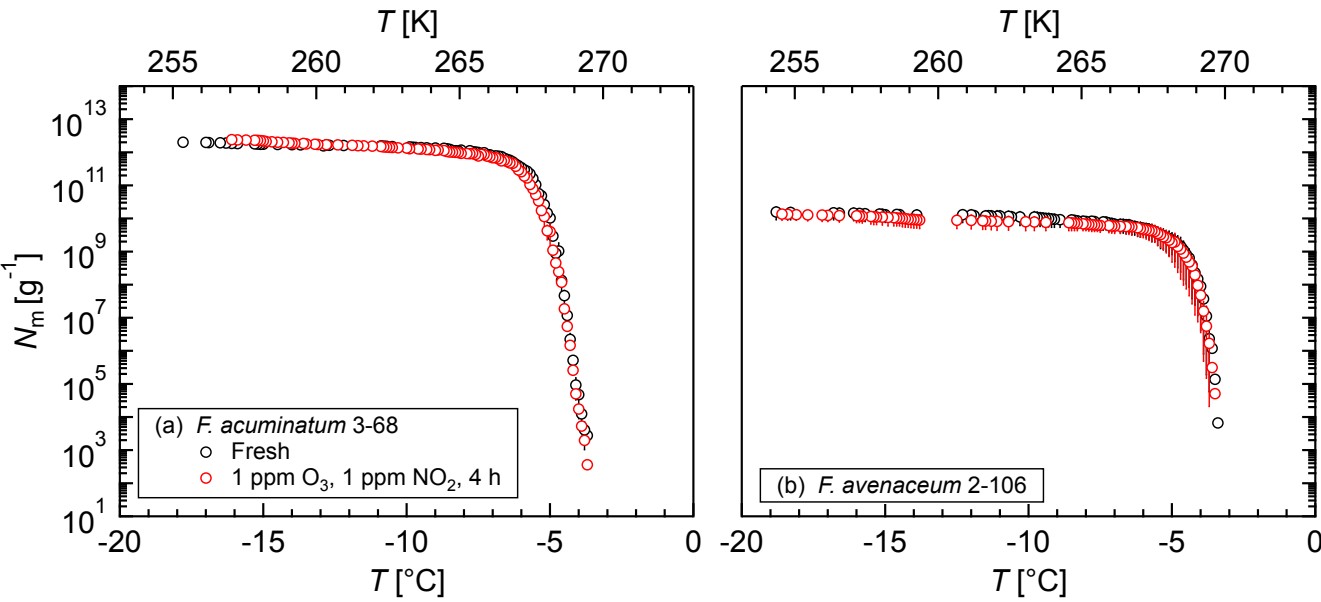

**Figure 3.** Exposure of aqueous extract from *Fusarium* to ozone and nitrogen dioxide: cumulative number of IN ($N_m$) per mass of mycelium plotted against the temperature ($T$) for (a) *F. acuminatum* 3-68 and (b) *F. avenaceum* 2-106; arithmetic mean values and standard error of two independent experiments with aqueous extracts from different fungal culture plates.

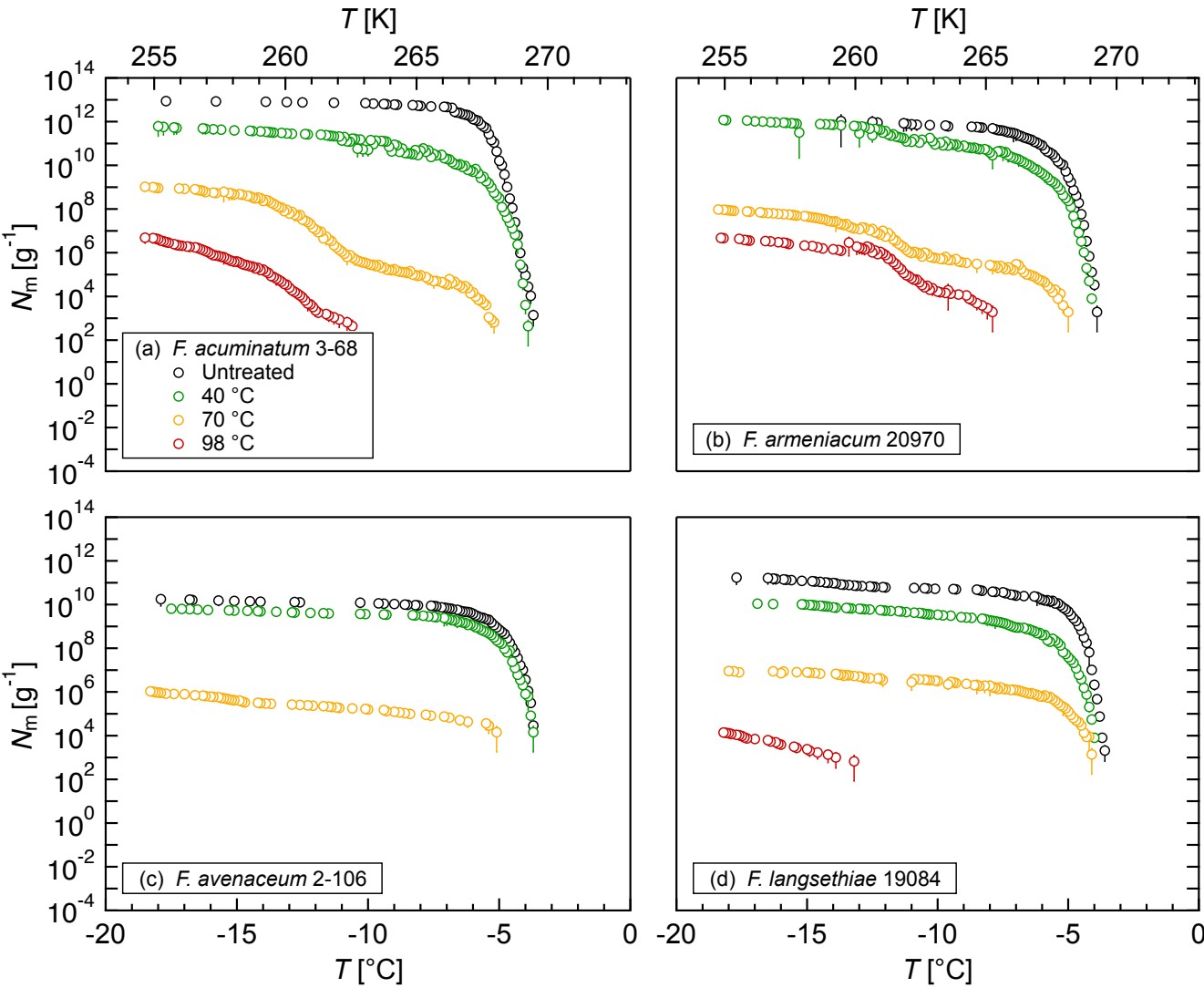

**Figure 4.** Effects of high temperatures on the ice nucleation activity of *Fusarium*: cumulative number of IN ($N_m$) per gram of mycelium plotted against the temperature ($T$) for (a) *F. acuminatum* 3-68, (b) *F. armeniacum* 20970, (c) *F. avenaceum* 2-106, and (d) *F. langsethiae* 19084. The error bars were calculated using the counting error and the Gaussian error propagation.

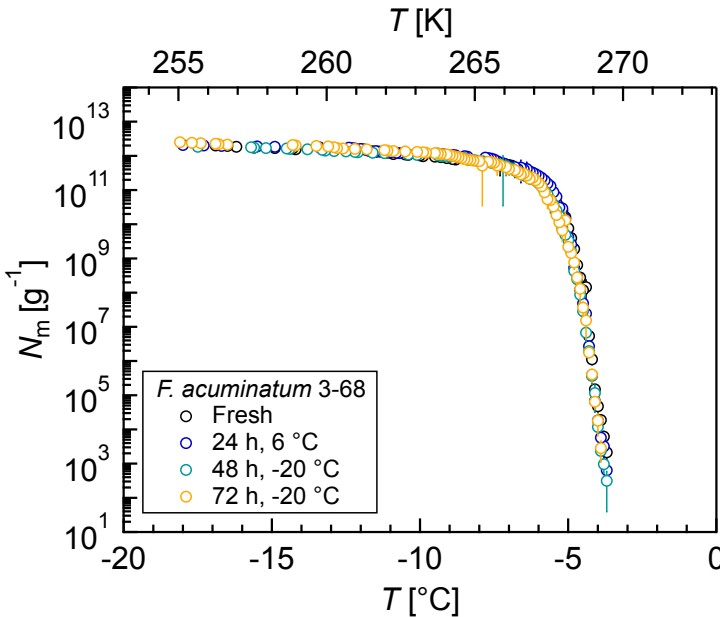

**Figure 5.** Effects of short-term storage and freeze-thaw cycles on the ice nucleation activity of *Fusarium acuminatum* 3-68: cumulative number of IN ($N_m$) per gram of mycelium plotted against the temperature ($T$). The same aqueous extract was measured immediately after preparation (black), after storage at 6 °C for 24 h (blue), after another 24 h stored at -20 °C (total 48 h; green), and after another 24 h stored at -20 °C (total 72 h; yellow). The error bars were calculated using the counting error and the Gaussian error propagation.

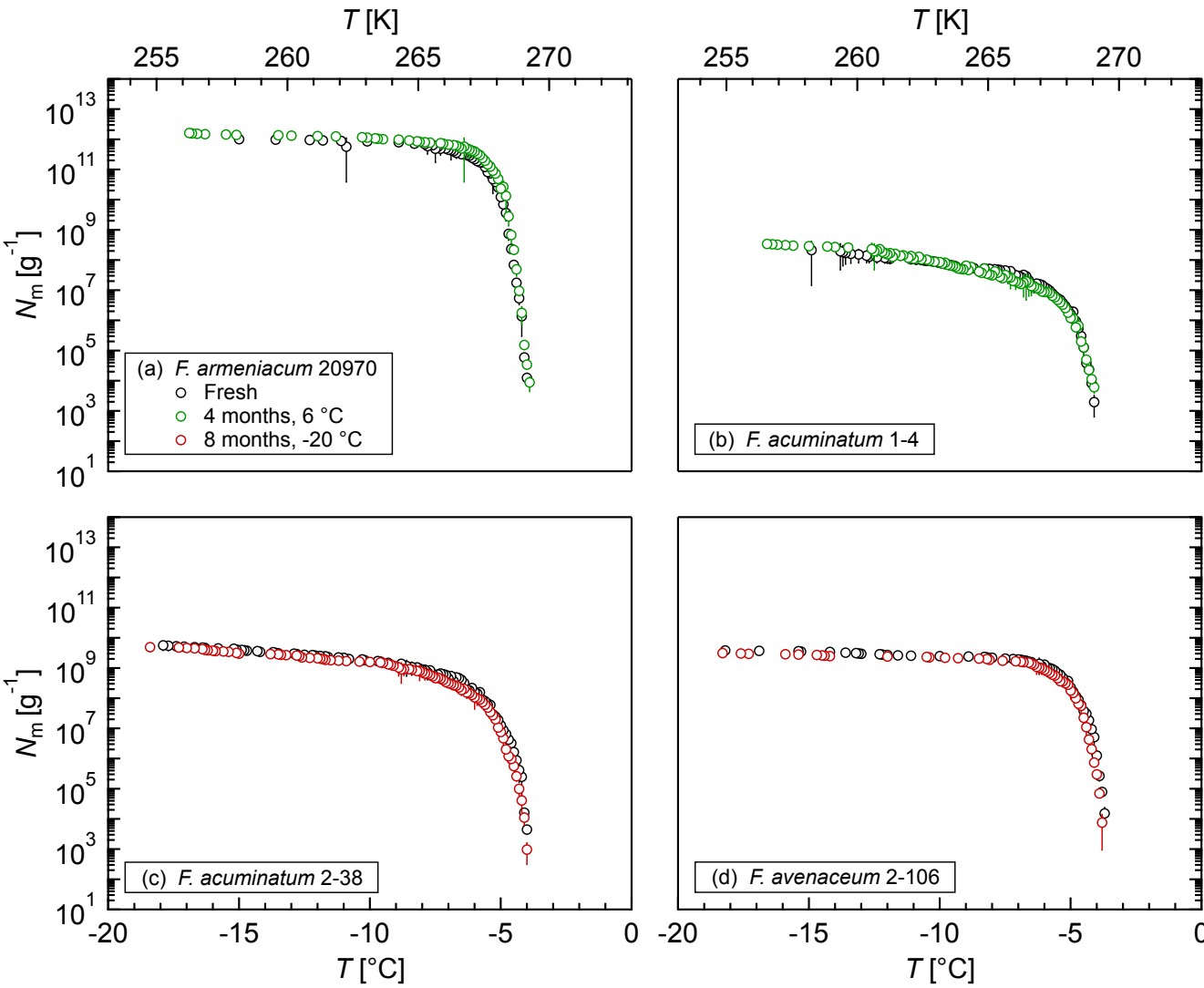

**Figure 6.** Effect of long-term storage on the ice nucleation activity of (a) *F. armeniacum* 20970, (b) *F. acuminatum* 1-4, (c) *F. acuminatum* 2-38, and (d) *F. avenaceum* 2-106: cumulative number of IN ($N_m$) per gram of mycelium against the temperature ($T$). The error bars were calculated using the counting error and the Gaussian error propagation.