# Peer review of "Macromolecular fungal ice nuclei in *Fusarium*: Effects of physical and chemical processing"

_Biogeosciences, 2019_

## Referee Comment (RC1) · Anonymous Referee #1 · 21 Aug 2019

General comments

The authors report on the ice nucleating ability of 112 strains from 65 different Fusarium species. Using standard, valid and well characterized drop freezing assays TINA and LINDA, the authors determined that 18/112 of the tests strains showed initial freezing temperatures of -3.5 to -11.2 oC and that 7/65 species had IN-"active" strains. The authors subsequently use some of the IN-active strains and submit them to physical processing by filtering and by refreezing the strains as well as to chemical processing by bubbling ozone and NO2 through the extract solution. Most of these processing experiments led to null results.

These results are certainly publishable, bring value to the field of bioaerosols and fit the scope of the journal. The manuscript can be accepted after the following comments

have been addressed, and likely after changes are made to improve the discussion of the results, including further quantification, blank data, reproducibility discussion and additional references for the context of the work.

Specific comments

1. Title: The title is misleading: the authors' conclusion is that 16% of the tested strains were ice active above -14 oC. I would argue that this percentage does not equate to "widespread". The authors also did substantial work with physical and chemical processing of their material which is not reflected in the title, but could be. For example, something along the lines of "Ice nucleation ability of 65 different Fusarium species: Effects of storage, size and chemical processing"

2. Abstract Why did the authors choose -14 oC as their threshold? This discussion should be added here in the abstract (and in the text as well).

The relevance of Fusarium should be explained in the abstract.

3. Introduction Lines 16-17: more recent references should be added, especially because of the mention of macromolecules. Also see review by (Knopf et al., 2018).

Lines 18-20: it would be important to mention nonetheless that recent work has made contributions to our understanding of IN and precipitation references by (Petters and Wright, 2015; Stopelli et al., 2015, 2017).

Lines 21-23: the 3+9 references could be better represented by explaining what each one has observed in one or two sentences each. This added discussion could help set the stage for the relevance of the work under review.

Lines 24-26: same comment as above in addition to this reference (Šantl-Temkiv et al., 2015)

Lines 28-30: when temperatures are reported, what fraction does it represented? The onset? 1%? Temperature when 50% of the droplets are frozen- T50? See (Vali, 2019)

Line 39: define the positive selective pressure for IN activity

It would be useful for the authors to discuss the mode of freezing investigated and why immersion freezing was used and what is its relevance.

Good overview of bioprecipitation. Great description of the evolutionary reasons for fungal species to be good ice nuclei.

4. Materials and methods In general, controls and filter blanks are missing from the data description and analysis and the authors are encouraged to show this data (perhaps in supplementary information) and to discuss this data. For example, what was the IN activity of the water background? What was the activity of the filter background? How did the backgrounds differ from LINDA to TINA?

It is clear that the authors used two techniques for their experiments, yet their discussion does not include any comparison plots or discussing the differences in the two instruments. Each figure (and Table S1) should also state which instrument was used to acquire the data.

Line 115: could the authors show the positive control data?

Lines 119: clarification: can the authors show their calculations here and are the data presented corrected for the freezing point depression or is the 0.5 C part of the overall uncertainty?

Additional experiment: dilution series of an active strain to see if the behaviour of the IN active material in solution is linear. I would argue that this experiment would be important to help support the seemingly accurate high freezing temperature data observed for certain strains, for example in Figures 3 and 4 and S1.

5. Results and discussion It is necessary for the authors to define their reported freezing temperatures. Are they the onset, the equivalent of one well freezing? If so, how do the authors address the recommendations of not using the onset addressed in (Polen et al., 2018)? Reporting freezing temperatures as T10 and T50 would be additionally

helpful.

Lines 141-144: could the authors offer a hypothesis to this lack of verifiability?

The hypothesis of proteinaceous material acting as IN is valid. What about polysaccharides? (Dreischmeier et al., 2017)

Line 166: was there any hypothesis associated with the selection of the strains presented in this section?

Size experiments should be compared to (Irish et al., 2019; Wilson et al., 2015) for example. In addition, the Wilson et al., Nature 2015 paper has a nm parameterization that the authors should include in their discussion of their values.

Lines 184-185: I do not understand how the authors arrived at this conclusion. According to figure 2, the majority of the IN activity was lost between 300 and 100 kDa. I would have concluded that the best IN are within that size, not smaller than 100 kDa. I agree with the authors nonetheless that there are still IN active material below 100 kDa, but not the most active.

For the discussion to flow, it would be important to explain in line 189 why Erickson came to that conclusion.

The null effect of chemical processing with O3 and NO2 was somewhat surprising. Based on (Borduas-Dedekind et al., 2019; Gute and Abbatt, 2018; Kunert et al., 2018), I would have expected to see oxidation of the proteinaceous material and thus decrease in IN ability. A discussion involving a hypothesis to the resistance of the strains to oxidation is warranted in light of these studies. Did the authors attempt to extend the exposure to longer times to force a reaction? On a pedantic note, I would argue that ozone exposure of 1 ppm over 4h is not equivalent to 200 ppb over 20h. The experiment was done while bubbling ozone into extracts and there are concentration effects to consider as well as the diffusion of the ozone could affect the chemistry. I would simply omit this sentence and just state the concentration with no mention of

equivalence.

Null results are difficult to present. To further substantiate the authors' conclusion, I would recommend that the authors show material that indeed reacted under their O3 and NO2 conditions. The authors did do a positive control (Lines 205-206) and showing that data would help further support their claim.

Finally, the storage effects were also null results, but did the authors also do a positive control? In any case, these results are very useful for the community.

Figure S1 arguably belongs in the text. The reproducibility between fungal culture plates is remarkably the largest change observed compared to other treatments such as O3 and NO2 exposure. A discussion relating this uncertainty to the other analyses would be important.

Report the weights of the mycelium measured gravimetrically (for example in Table S1).

Is there value in considering the work in the context of food science and cryogenic food storage? Is it more likely that these strains be found in food or in the atmosphere?

Table S1 should present quantitative details. The authors should specify what their criteria is for "IN-active" strains. 1/96 wells? Onset? Temperature range? It would also be useful to add a fourth column with the freezing temperatures (T10 or T50 or T90). Did the authors consider making a parameterization with their data as an upper limit of IN activity of Fusarium species?

6. Conclusion I would revise the statement on line 226 to say that the most IN-active components were actually between 300-100 kDa, but that IN activity still remained smaller than 100 kDa.

Technical comments - The authors use upper case Nm which is arguably inconsistent with the literature using lower case nm. See Wex et al., ACP, 2015 - Line 14: "impact" should be replaced by "implication", since the authors did not quantify the water cycle

or the climate in their experiments. - The short summary is very good indeed! (although I would recommend changing the statement to 300 kDa, rather than 100 kDa.)

References: Borduas-Dedekind, N., Ossola, R., David, R. O., Boynton, L. S., Weichlinger, V., Kanji, Z. A. and McNeill, K.: Photomineralization mechanism changes the ability of dissolved organic matter to activate cloud droplets and to nucleate ice crystals, Atmospheric Chemistry and Physics Discussions, 1–27, doi:https://doi.org/10.5194/acp-2019-427, 2019.

Dreischmeier, K., Budke, C., Wiehemeier, L., Kottke, T. and Koop, T.: Boreal pollen contain ice-nucleating as well as ice-binding "antifreeze" polysaccharides, Sci. Rep., 7, doi:10.1038/srep41890, 2017.

Gute, E. and Abbatt, J. P. D.: Oxidative Processing Lowers the Ice Nucleation Activity of Birch and Alder Pollen, Geophysical Research Letters, 45(3), 1647–1653, doi:10.1002/2017GL076357, 2018.

Irish, V. E., Hanna, S. J., Xi, Y., Boyer, M., Polishchuk, E., Ahmed, M., Chen, J., Abbatt, J. P. D., Gosselin, M., Chang, R., Miller, L. A. and Bertram, A. K.: Revisiting properties and concentrations of ice-nucleating particles in the sea surface microlayer and bulk seawater in the Canadian Arctic during summer, Atmospheric Chemistry and Physics, 19(11), 7775–7787, doi:https://doi.org/10.5194/acp-19-7775-2019, 2019.

Knopf, D. A., Alpert, P. A. and Wang, B.: The role of organic aerosol in atmospheric ice nucleation: A review, ACS Earth Space Chem., 2(3), 168–202, doi:10.1021/acsearthspacechem.7b00120, 2018.

Kunert, A. T., Lamneck, M., Helleis, F., Pöschl, U., Pöhlker, M. L. and Fröhlich-Nowoisky, J.: Twin-plate Ice Nucleation Assay (TINA) with infrared detection for high-throughput droplet freezing experiments with biological ice nuclei in laboratory and field samples, Atmospheric Measurement Techniques, 11(11), 6327–6337, doi:https://doi.org/10.5194/amt-11-6327-2018, 2018.

Petters, M. D. and Wright, T. P.: Revisiting ice nucleation from precipitation samples, Geophysical Research Letters, 42(20), 8758–8766, doi:10.1002/2015GL065733, 2015.

Polen, M., Brubaker, T., Somers, J. and Sullivan, R. C.: Cleaning up our water: reducing interferences from nonhomogeneous freezing of "pure" water in droplet freezing assays of ice-nucleating particles, Atmospheric Measurement Techniques, 11(9), 5315–5334, doi:https://doi.org/10.5194/amt-11-5315-2018, 2018.

Šantl-Temkiv, T., Sahyoun, M., Finster, K., Hartmann, S., Augustin-Bauditz, S., Stratmann, F., Wex, H., Clauss, T., Nielsen, N. W., Sørensen, J. H., Korsholm, U. S., Wick, L. Y. and Karlson, U. G.: Characterization of airborne ice-nucleation-active bacteria and bacterial fragments, Atmospheric Environment, 109, 105–117, doi:10.1016/j.atmosenv.2015.02.060, 2015.

Stopelli, E., Conen, F., Morris, C. E., Herrmann, E., Bukowiecki, N. and Alewell, C.: Ice nucleation active particles are efficiently removed by precipitating clouds, Scientific Reports, 5, 16433, doi:10.1038/srep16433, 2015.

Stopelli, E., Conen, F., Guilbaud, C., Zopfi, J., Alewell, C. and Morris, C. E.: Ice nucleators, bacterial cells and Pseudomonas syringae in precipitation at Jungfraujoch, Biogeosciences, 14(5), 1189–1196, doi:10.5194/bg-14-1189-2017, 2017.

Vali, G.: Revisiting the differential freezing nucleus spectra derived from drop-freezing experiments: methods of calculation, applications, and confidence limits, Atmos. Meas. Tech., 12(2), 1219–1231, doi:https://doi.org/10.5194/amt-12-1219-2019, 2019.

Wilson, T. W., Ladino, L. A., Alpert, P. A., Breckels, M. N., Brooks, I. M., Browse, J., Burrows, S. M., Carslaw, K. S., Huffman, J. A., Judd, C., Kilthau, W. P., Mason, R. H., McFiggans, G., Miller, L. A., Najera, J. J., Polishchuk, E., Rae, S., Schiller, C. L., Si, M., Temprado, J. V., Whale, T. F., Wong, J. P. S., Wurl, O., Yakobi-Hancock, J., Abbatt, J. P. D., Aller, J. Y., Bertram, A. K., Knopf, D. A. and Murray, B. J.: A marine biogenic source

of atmospheric ice-nucleating particles, Nature, 525(7568), 234–238, 2015.

---

## Referee Comment (RC2) · Anonymous Referee #2 · 10 Sep 2019

General Comments:

The authors report on Fusarium species ice nucleation activity utilizing two standard, well established methods. 112 strains from 65 species were analyzed using LINDA and TINA and identified 18 strains with initial ice nucleation activity between -3.5 and -12°C. It was further demonstrated that freeze-thaw cycles did not impact ice nucleation activity and filtration of the samples suggests that the proteinaceous compound responsible for Fusarium ice nucleation forms cell-free aggregates whose nucleation efficacy is impacted by size. Additional ozone and NO2 studies were done to further evaluate the stability of the compounds in the atmosphere and showed no variation from untreated samples.

This work provides valuable insight into the biodiversity of ice nucleation active Fusarium species and aids in the understanding of ice nucleating particles and bioaerosols as a whole. This work is aligned with the scope of the journal and can be accepted after the noted changes have been addressed.

Specific comments:

Abstract: Indicate the biological relevance of Fusarium and its ice nucleation activity. This is discussed well in the introduction but will help to bridge the first few sentences of the abstract.

Methods 2.1: How were the initial samples obtained? Could their original environment (crop vs. airborne, etc.) shed light on IN frequency?

Line 21: Additional, more recent, studies have contributed to this understanding of IN as well. (Failor et. al. 2017, Hanlon et al. 2017, Stopelli et al. 2017, 2015, Joly et al. 2014).

Line 24-6: Failor et al. (2017) further expanded on known gammaproteobacteria IN.

Line 118: Was the range of incubation times necessary to reach a specified optical density? If so, that indication would be useful. If not, elaborate of reasoning for the times.

Line 119: Be specific for the 0.5°C freezing point depression. Is it 0.5°C or 0.5±x°C.

Results 3.1: This would be an interesting point to note the original sampling locations for the various strains and could further demonstrate the cosmopolitan nature of these IN-active species should any tends be identified.

Lines 154-5: This is a risky assumption to make. Prior to the Failor et al. study, all bacterial IN were thought to be proteinaceous. Exposing a selection of the species to high heat could support this claim.

Lines 184-6: With the drastic decrease in activity after the 300,000 MWCO filter and then again after 100,000, could the protein not be larger, but when damaged or broken

still retains some ice nucleation activity?

Lines 195-6: Why would single proteins in the atmosphere be unlikely? Please elaborate on this statement.

Line 216: Change "...and the fungus could safe energy." to "...and the fungus could save energy.".

Figure 1. Inclusion of the positive control SnoMax curve would be beneficial here. Any incidence of spontaneous freezing of the negative control should also be noted (if any occurred with the methods you used).

Figure 3. You note in the text that SnoMax has been shown to decrease after exposure. Did you see this same result, or did you not use SnoMax because of this interaction?

References:

Failor, K.C., Schmale III, D.G., Vinatzer, B.A., and Monteil, C.L.: Ice nucleation active bacteria in precipitation are genetically diverse and nucleate ice by employing different mechanisms, ISMEJ, 11(12), 2740–2753, doi:10.1038/ismej.2017.124, 2017.

Hanlon, R., Powers, C., Failor, K.C., Monteil, C.L., Vinatzer, B.A., and Schmale III, D.G.: Microbial ice nucleators scavenged from the atmosphere during simulated rain events, Atmospheric Environment, 163, 182-189, doi:10.1016/j.atmosenv.2017.05.030, 2017.

Joly, M., Amato, P., Deguillaume, L., Monier, M., Hoose, C., and Delort, A.-M.: Quantification of ice nuclei active at near 0°C temperatures in low-altitude clouds at the Puy de Dôme atmospheric station, Atmospheric Chemistry and Physics, 14, 8185-8195, doi:10.5194/acp-14-8185-2014, 2014.

Stopelli, E., Conen, F., Morris, C. E., Herrmann, E., Bukowiecki, N. and Alewell, C.: Ice nucleation active particles are efficiently removed by precipitating clouds, Scientific Reports, 5, 16433, doi:10.1038/srep16433, 2015.

Stopelli, E., Conen, F., Guilbaud, C., Zopfi, J., Alewell, C. and Morris, C. E.: Ice nucleators, bacterial cells and Pseudomonas syringae in precipitation at Jungfraujoch, Biogeosciences, 14(5), 1189–1196, doi:10.5194/bg-14-1189-2017, 2017.

———————————————————

---

## Referee Comment (RC3) · Anonymous Referee #3 · 20 Sep 2019

This is a very well-written manuscript concerning ice nucleation active fungal species. The paper highlights the high ice nucleation activity of Fusarium which compares to the best ice nucleating particles known so far. A main finding of this study is that filtration experiments suggest that the single cell-free Fusarium is smaller than 100 kDa. This is indeed very interesting and I wonder that the authors do not use the nomenclature of their own paper (Pummer et al., 2015) i.e. ice nucleating macromolecules (INM). Indeed, water-soluble INMs have also been observed on many other primary biological aerosol particles (PABP) such as leaves, bark, pollen (Felgitsch et al., 2018), algae (Tesson et al., 2018), and bacteria (Failor et al., 2017). The sizes of these INM should be compared among each other, e.g. in a table. The same is true for the chemical composition and for the stability against oxidation. Also for other PBAPs, proteins

and polysaccharides have been found as main components of INM and their stability is extraordinary as well. I also wonder if the authors have carried out heating experiments in order to destroy the ice nucleation activity of the proteins. Eventually, the heating was not successful due to the stability of INMs which would be important information since many colleagues use heating experiments to prove or unprove the presence of PBAP-INPs. Altogether, this is a very valuable manuscript which should be published after some minor corrections.

Comment

The abbreviation "IN" has been used in a confusing way. In the text it means "ice nuclei" but also means "ice nucleation" and "ice nucleating". I recommend using "INP" for "ice nucleating particles" and write the full words in all other cases.

References

Failor, K.C., Schmale III, D.G., Vinatzer, B.A., Monteil C.L.: Ice nucleation active bacteria in precipitation are genetically diverse and nucleate ice by employing different mechanisms, The ISME journal 11, 12, 2740 – 2753, 2017.

Felgitsch, L., Baloh, P., Burkart, J., Mayr, M., Momken, M. E., Seifried, T. M., Winkler, P., Schmale III, D. G., and Grothe, H.: Birch leaves and branches as a source of ice-nucleating macromolecules, Atmospheric Chemistry and Physics, 18, 16063 – 16079, 2018.

Pummer, B. G., Budke, C., Augustin-Bauditz, S., Niedermeier, D., Felgitsch, L., Kampf, C., Huber, R., Liedl, K., Loerting, T., Moschen, T., Schauperl, M., Tollinger, M., Morris, C., Wex, H., Grothe, H., Pöschl, U., Koop, T., Fröhlich-Nowoisky, J.: Ice nucleation by water-soluble macromolecules, Atmospheric Chemistry and Physics, 15, 4077 – 4091, 2015.

Tesson, S. V. M., Šantl-Temkiv, T.: Ice Nucleation Activity and Aeolian Dispersal Success in Airborne and Aquatic Microalgae, Frontiers in Microbiology, 9, 2681, 2018,

https://doi.org/10.3389/fmicb.2018.02681

---

## Author Comment (AC1) · 21 Oct 2019

**MS bg-2019-276, Kunert et al.: Highly active and stable fungal ice nuclei are widespread among Fusarium species**

We thank referee #1 for his/her comments, questions, and suggestions, which have been taken into account upon revision of our manuscript. The comments and our answers are listed below (referee's comments marked with blue letters).

1. Title:

Referee comment: The title is misleading: the authors' conclusion is that 16% of the tested strains were ice active above -14 °C. I would argue that this percentage does not equate to "widespread". The authors also did substantial work with physical and chemical processing of their material which is not reflected in the title, but could be. For example, something along the lines of "Ice nucleation ability of 65 different Fusarium species: Effects of storage, size and chemical processing"

Author's response: We changed the title to: "Macromolecular fungal ice nuclei in *Fusarium*: Effects of physical and chemical processing" and modified the corresponding parts in the manuscript accordingly.

2. Abstract:

Referee comment: Why did the authors choose -14 °C as their threshold? This discussion should be added here in the abstract (and in the text as well).

Author's response: We thank the referee for pointing this out. We actually meant -12 °C, and we realized that we had a typing error here, which we corrected now.

As *Fusarium* nucleates in a broad temperature range between -1 and -9 °C (Hasegawa et al., 1994; Humphreys et al., 2001; Pouleur et al., 1992; Richard et al., 1996; Tsumuki et al., 1992; Tsumuki and Konno, 1994), and as the water background of LINDA started to freeze at -14 °C, we set the threshold to -12 °C.

Referee comment: The relevance of Fusarium should be explained in the abstract.

Author's response: We thank the referee for this suggestion and included the following sentences in the abstract: "Ice nucleation activity in fungi was first discovered in the cosmopolitan genus *Fusarium*, which is widespread in soil and plants, has been found in atmospheric aerosol and cloud water samples, and can be regarded as the best studied IN-active fungus."

Moreover, we modified the following sentences: "The frequency and distribution of ice nucleation activity within *Fusarium*, however, remains elusive. Here, we tested more than 100 strains from 65 different *Fusarium* species for ice nucleation activity."

3. Introduction:

Referee comment: Lines 16-17: more recent references should be added, especially because of the mention of macromolecules. Also see review by (Knopf et al., 2018).

Author's response: We thank the referee for this comment and included further references in our manuscript.

Referee comment: Lines 18-20: it would be important to mention nonetheless that recent work has made contributions to our understanding of IN and precipitation references by (Petters and Wright, 2015; Stopelli et al., 2015, 2017).

Author's response: We thank the referee for this remark and added the references to our manuscript.

Referee comment: Lines 21-23: the 3+9 references could be better represented by explaining what each one has observed in one or two sentences each. This added discussion could help set the stage for the relevance of the work under review.
Author's response: The discussion of each reference in one or two sentences each would result in a very long introduction with a review character, especially as referee #2 suggested to add additional references here. Adding a detailed discussion of the different types of biological ice nuclei at this point goes beyond the scope and focus of this manuscript and could lead to confusion of the readers. Instead, some of the references are discussed in more detail in the results and discussion section.

Referee comment: Lines 24-26: same comment as above in addition to this reference (Šantl-Temkiv et al., 2015)
Author's response: We included the suggested reference as well as the reference, which was suggested by referee #2 (Failor et al., 2017), but we prefer to not extend the bacterial IN part of the introduction as the focus of the manuscript should be on fungi, particularly *Fusarium*.

We modified the following sentences: "The best characterized biological IN are common plant-associated bacteria of the genera *Pseudomonas*, *Pantoea*, and *Xanthomonas* (Garnham et al., 2011; Govindarajan and Lindow, 1988; Graether and Jia, 2001; Green and Warren, 1985; Hill et al., 2014; Kim et al., 1987; Ling et al., 2018; Schmid et al., 1997; Wolber et al., 1986), and recently, an ice nucleation-active (IN-active) *Lysinibacillus* was found (Failor et al., 2017). The first identified IN-active fungi were strains of the genus *Fusarium* (Hasegawa et al., 1994, Pouleur et al., 1992, Richard et al., 1996, Tsumuki et al., 1992)."

Referee comment: Lines 28-30: when temperatures are reported, what fraction does it represented? The onset? 1%? Temperature when 50% of the droplets are frozen- T50? See (Vali, 2019)
Author's response: As mentioned in Line 28, the temperatures are reported as initial freezing temperatures, which corresponds to the onset freezing temperature: "To date, a few more fungal genera with varying initial freezing temperatures such as *Isaria farinosa* ($\sim$ -4 °C), *Mortierella alpina* ($\sim$ -5 °C), *Puccinia species* (-4 °C to -8 °C), and *Sarocladium* (formerly named *Acremonium*) *implicatum* ($\sim$ -9 °C) have been identified as IN-active (Fröhlich-Nowoisky et al., 2015; Huffman et al., 2013; Morris et al., 2013; Richard et al., 1996)".

Referee comment: Line 39: define the positive selective pressure for IN activity
Author's response: We thank the referee for pointing out the ambiguity of our statement. For clarification, we modified the sentence: "While the factors for a positive selective pressure for ice nucleation activity in *Fusarium* and other fungi have not been directly identified, an ecological advantage of initiating ice formation is easily conceivable." For example, the bioprecipitation feedback cycle can be such a factor, which is discussed in more detail later (Lines 47-49).

Referee comment: It would be useful for the authors to discuss the mode of freezing investigated and why immersion freezing was used and what is its relevance.
Author's response: The droplet freezing assays, which were used in this study, all measure ice nucleation activity in the immersion freezing mode, where the IN is contained inside a liquid droplet when initiating freezing. Biological IN are often proteins, which are surrounded by a hydration shell, so the immersion freezing mode is suitable for biological IN. Thus, the most common techniques to study biological IN are droplet freezing assays (Després et al., 2012; Hoose and Möhler, 2012).

To avoid misunderstanding, we modified the sentence: "Ice nuclei of selected *Fusarium* species
were further analyzed in immersion freezing mode using the high-throughput Twin-plate Ice
Nucleation Assay (TINA) (Kunert et al., 2018)."

Referee comment: Good overview of bioprecipitation. Great description of the evolutionary
reasons for fungal species to be good ice nuclei.
Author's response: We thank the referee for this comment.

4.   Materials and Methods
Referee comment: In general, controls and filter blanks are missing from the data description
and analysis and the authors are encouraged to show this data (perhaps in supplementary
information) and to discuss this data. For example, what was the IN activity of the water
background? What was the activity of the filter background? How did the backgrounds differ
from LINDA to TINA?
Author's response: We added the information about the negative controls and included the
following sentences in the manuscript:

For the thermal cycler: "Aliquots of uninoculated DPY broth were used as negative controls,
which did not freeze in the investigated temperature interval."

For LINDA experiments: "As a negative control, a 0.9 % NaCl solution was added to three
uninoculated agar plates, and the freezing started below -14 °C."

For TINA experiments: "Pure water samples (0.1 µm filtered) served as a negative control for
each experiment. These did not freeze in the observed temperature interval."

Referee comment: It is clear that the authors used two techniques for their experiments, yet
their discussion does not include any comparison plots or discussing the differences in the two
instruments. Each figure (and Table S1) should also state which instrument was used to acquire
the data.
Author's response: As described in Lines 104-105, the initial screening was performed with
two independent droplet freezing assays in two laboratories. Strains of the USDA-
ARS/Michigan State University were screened with a thermal cycler as described in Fröhlich-
Nowoisky et al. (2015) (Lines 106-108). Strains from the Schmale laboratory at Virginia Tech
and strains from the Kansas State University Teaching Collection were screened with LINDA
(Lines 111-113). Table S1 provides a summary of all tested strains, the strain collection they
originate from, and the results of the screening. Table 1 shows the mean freezing temperatures
for the positively tested species. All further analyses were performed with TINA.

Referee comment: Line 115: could the authors show the positive control data?
Author's response: We included the following sentence in section 2.3: "The freezing
temperatures ranged from -3.46 °C to -4.58 °C."

Referee comment: Lines 119: clarification: can the authors show their calculations here and are
the data presented corrected for the freezing point depression or is the 0.5 C part of the overall
uncertainty?
Author's response: We added the calculations to the supplementary information.

The data presented here were not corrected for the freezing point depression as highly
concentrated *Fusarium* extracts were used for the initial screening. Thus, we cannot exclude
that the high concentration of *Fusarium* IN compensates the effect of NaCl on the freezing temperature. We added this information in the manuscript: "We cannot exclude, however, that the high concentration of IN compensates the effect of NaCl on the freezing temperature. This is supported by the investigations of Stopelli et al. (2014), who did not find a systematic suppression of freezing at this salt concentration in LINDA experiments."

Referee comment: Additional experiment: dilution series of an active strain to see if the behaviour of the IN active material in solution is linear. I would argue that this experiment would be important to help support the seemingly accurate high freezing temperature data observed for certain strains, for example in Figures 3 and 4 and S1.
Author's response: All samples, which were analyzed with TINA, were measured in a dilution series. We described this in Lines 121-123: "The aqueous extracts were serially diluted 10-fold with pure water by a liquid handling station (epMotion ep5073, Eppendorf, Hamburg, Germany), and 96 droplets (3 µL) were tested per dilution with a continuous cooling rate of 1 °C min$^{-1}$ from 0 °C to -20 °C."

For clarification, we optimized the sentences: "The aqueous extract was serially diluted 10-fold with pure water by a liquid handling station (epMotion ep5073, Eppendorf, Hamburg, Germany) to a dilution where droplets remained liquid in the investigated temperature interval. Of each dilution, 96 droplets (3 µL) were tested with a continuous cooling rate of 1 °C min$^{-1}$ from 0 °C to -20 °C."

5.  Results and Discussion
Referee comment: It is necessary for the authors to define their reported freezing temperatures. Are they the onset, the equivalent of one well freezing? If so, how do the authors address the recommendations of not using the onset addressed in (Polen et al., 2018)? Reporting freezing temperatures as T10 and T50 would be additionally helpful.
Author's response: Except for the initial screening, we always report the initial freezing temperatures ($T_i$) for our measurements, which is equivalent to the onset. We first reported the freezing temperatures for the initial screening as initial freezing temperatures, but we actually meant mean freezing temperatures.

We replaced "initial" by "mean" several times in the text, where we talk about the initial screening.

Referee comment: Lines 141-144: could the authors offer a hypothesis to this lack of verifiability?
Author's response: The fungal culture plates, which were used for the initial screening, could not be used for the measurements with TINA, as different laboratories were involved in this study. Moreover, it is well known that some *Fusarium* species can reduce or lose their IN activity after several subcultures (Pummer et al., 2013; Tsumuki et al., 1995). We discussed this in Lines 156-159 in the manuscript: "It is known that *Fusarium* can regulate the gene expression for IN production depending on environmental conditions such as nutrient availability (Richard et al., 1996), and some *Fusarium* species reduce or lose their IN activity after several subcultures (Pummer et al., 2013; Tsumuki et al., 1995)."

Referee comment: The hypothesis of proteinaceous material acting as IN is valid. What about polysaccharides? (Dreischmeier et al., 2017)
Author's response: We cannot exclude that polysaccharides are involved in the ice nucleation of *Fusarium*. To our knowledge, however, there is no published study showing that polysaccharides are involved in the ice nucleation activity of *Fusarium*.

We discussed a potential role in section 3.3: "The remaining activity after the 98 °C treatment,
however, could indicate that post-translational modifications like glycosylation and therefore
polysaccharides could play a role in the ice nucleation activity of *Fusarium*. Further systematic
studies including chemical analyses are needed for elucidation."

We included the following sentence in the conclusion: "An involvement of polysaccharides,
however, cannot be excluded."

Referee comment: Line 166: was there any hypothesis associated with the selection of the
strains presented in this section?
Author's response: Not all *Fusarium* strains were available for the experiments with TINA, as
the initial screening was performed in different laboratories. But we tried to cover as many
different species as possible and selected species, which were long known for ice nucleation
activity (*F. acuminatum*, *F. avenaceum*) as well as all the newly identified species.

For clarification, we included this information in section 2.3: "Ice nuclei of selected *Fusarium*
species, which were long known for ice nucleation activity (*F. acuminatum*, *F. avenaceum*) as
well as all the newly identified species, were further analyzed in immersion freezing mode
using the high-throughput Twin-plate Ice Nucleation Assay (TINA) (Kunert et al., 2018)."

Referee comment: Size experiments should be compared to (Irish et al., 2019; Wilson et al.,
2015) for example. In addition, the Wilson et al., Nature 2015 paper has a nm parameterization
that the authors should include in their discussion of their values.
Author's response: We included the following sentence: "Moreover, biological INMs smaller
than 200 nm were also found in various organisms e.g., other fungi (Fröhlich-Nowoisky et al.,
2015; Pummer et al., 2015), leaves, bark, and pollen from birch trees (*Betula* spp.) (Felgitsch
et al.,2018; Pummer et al., 2012), leaf litter (Schnell and Vali, 1973), some microalgae (Tesson
and Šantl-Temkiv, 2018), strains of *Lysinibacillus* (Failor et al., 2017), and biological particles
in the sea surface microlayer (Irish et al., 2019; Wilson et al., 2015)."

Referee comment: Lines 184-185: I do not understand how the authors arrived at this
conclusion. According to figure 2, the majority of the IN activity was lost between 300 and 100
kDa. I would have concluded that the best IN are within that size, not smaller than 100 kDa. I
agree with the authors nonetheless that there are still IN active material below 100 kDa, but not
the most active.
Author's response: As IN were found in all size fractions, we concluded that *Fusarium* IN are
likely single proteins smaller than 100 kDa, which can agglomerate to large protein complexes
in solution. We did not claim that the single proteins smaller than 100 kDa are the most active
ones. Lines 184-185: "We hypothesize that *Fusarium* IN are single proteins smaller than 100
kDa, which agglomerate to large protein complexes in solution."

As explained in Lines 177-178, filtration through a 300 000 MWCO filter unit decreased the
cumulative number of IN per gram of mycelium about 50 % to 75 %. Further filtration through
a 100 000 MWCO filter unit reduced the IN number to less than 1 % of the initial concentration
(Lines 180-181). So, the majority was lost upon 300 000 MWCO filtration, which were the
most efficient IN nucleating at the highest temperatures.

Referee comment: For the discussion to flow, it would be important to explain in line 189 why
Erickson came to that conclusion.
Author's response: We changed the sentence to: "Erickson (2009) determined the size of
proteins based on theoretical calculations. As the interior of proteins is closely packed with no substantial holes and almost no water molecules inside, proteins are rigid structures with
approximately the same density ($\sim$1.37 g cm$^{-1}$). Assuming the protein as a smooth spherical
particle, the minimum diameter of the INM would be smaller than 6.1 nm".
Referee comment: The null effect of chemical processing with O3 and NO2 was somewhat
surprising. Based on (Borduas-Dedekind et al., 2019; Gute and Abbatt, 2018; Kunert et al.,
2018), I would have expected to see oxidation of the proteinaceous material and thus decrease
in IN ability. A discussion involving a hypothesis to the resistance of the strains to oxidation is
warranted in light of these studies. Did the authors attempt to extend the exposure to longer
times to force a reaction? On a pedantic note, I would argue that ozone exposure of 1 ppm over
4h is not equivalent to 200 ppb over 20h. The experiment was done while bubbling ozone into
extracts and there are concentration effects to consider as well as the diffusion of the ozone
could affect the chemistry. I would simply omit this sentence and just state the concentration
with no mention of equivalence.
Author's response: Based on our results, we cannot exclude that post-translational
modifications of the *Fusarium* IN protein occurred during oxidation. These potential
modifications do not seem to influence the ice nucleation activity of the protein. For example,
they could be in parts of the protein, which are not involved in the nucleation process. We agree
with the referee that further investigations are necessary, and we will consider these
experiments for future studies.
Moreover, we included the suggested references in the manuscript and extended the following
sentence: "This is in contrast to other biological IN e.g., bacterial IN (Snomax$^{\textregistered}$) (Kunert et al.,
2018), birch and alder pollen (Gute and Abbatt, 2018), and dissolved organic matter (Borduas-
Dedekind et al., 2019), where exposure to oxidizing agents reduced the IN activity."
We deleted the statement and modified the following sentence: "Briefly, a mixture of 1 ppm O$_3$
and 1 ppm NO$_2$ was bubbled through 1 mL aliquots of aqueous extract for 4 h, and the IN
concentration was determined using TINA."
Referee comment: Null results are difficult to present. To further substantiate the authors'
conclusion, I would recommend that the authors show material that indeed reacted under their
O3 and NO2 conditions. The authors did do a positive control (Lines 205-206) and showing
that data would help further support their claim.
Author's response: As the focus of this study is on fungal IN of *Fusarium*, we did not use
Snomax in any of the experiments. As described in the manuscript (Lines 205-206), we found
a reduction of IN activity upon exposure to O$_3$ and NO$_2$ for Snomax in a previous study (Kunert
et al., 2018).
Referee comment: Finally, the storage effects were also null results, but did the authors also do
a positive control? In any case, these results are very useful for the community.
Author's response: We could not include a positive control in our storage tests as a suitable
control for such experiments was not available. We agree that further IN should be tested for
effects of storage.
Referee comment: Figure S1 arguably belongs in the text. The reproducibility between fungal
culture plates is remarkably the largest change observed compared to other treatments such as
O3 and NO2 exposure. A discussion relating this uncertainty to the other analyses would be
important.
Author's response: The data in Figure S1 were obtained from three different fungal culture
plates, whereas the exposure experiments were performed with the same aqueous extract of the particular fungal species. The variability of measurement with individual fungal culture plates is higher than measurements of the same aqueous extract, as the differences did not result from the measurements themselves but rather from the fact that we investigated biological samples.

Referee comment: Report the weights of the mycelium measured gravimetrically (for example in Table S1).

Author's response: Table S1 shows the results of the initial screening, which was performed with two different droplet freezing assays, first a thermal cycler and second the LINDA instrument (section 2.3). For the thermal cycler, mycelium was picked and directly transferred into 96-well PCR plates (Lines 108-110), and for LINDA, 0.9 % NaCl solution was added to the fungal culture plates, which were scraped afterwards to obtain a suspension of mycelium and spores (Lines 80-82). As the initial screening was only a yes or no test, it was not deemed necessary to determine the weight of the mycelium.

Referee comment: Is there value in considering the work in the context of food science and cryogenic food storage? Is it more likely that these strains be found in food or in the atmosphere?

Author's response: *Fusarium* species are frequently associated with plant material (Leslie and Summerell, 2006), including many food types, and some of the strains used in the current study were initially isolated from plants. Thus, IN from such fungi could be important in food response to freezing temperatures, which could be worth future investigation. Considering the work in the context of food science and cryogenic food storage, however, would be outside the scope of this manuscript, in which we focus on atmospheric aspects of ice nucleation activity in *Fusarium*.

Referee comment: Table S1 should present quantitative details. The authors should specify what their criteria is for "IN-active" strains. 1/96 wells? Onset? Temperature range? It would also be useful to add a fourth column with the freezing temperatures (T10 or T50 or T90). Did the authors consider making a parameterization with their data as an upper limit of IN activity of Fusarium species?

Author's response: For the initial screening using the thermal cycler, up to seven droplets were investigated for each sample. If the sample was IN-active, all droplets froze in the investigated temperature interval. We included the following sentence: "Up to seven droplets were measured for each sample, and the mean freezing temperature was calculated."

For the initial screening with LINDA, three droplets were investigated for each sample, which was described in the manuscript in Lines 113-114: "Aliquots of 200 µL of each aqueous extract were transferred to three separate 500 µL tubes and placed on ice for 1 h prior to the LINDA experiments." If the sample was IN-active, all droplets froze in the investigated temperature interval. For clarification, we included the following sentence: "The mean freezing temperature for three droplets was calculated."

The suggested fourth column would correspond to Table 1, which already provides more details about the mean freezing temperatures of the initial screening.

We thank the referee for this suggestion, and we will consider a parameterization in a future study.

6. Conclusion

Referee comment: I would revise the statement on line 226 to say that the most IN-active
components were actually between 300-100 kDa, but that IN activity still remained smaller
than 100 kDa.
Author's response: As described above, the most IN-active components were larger than 300
kDa, and we hypothesize that these are aggregates consisting of individual proteins smaller than
100 kDa.
Technical comments
Referee comment: The authors use upper case Nm which is arguably inconsistent with the
literature using lower case nm. See Wex et al., ACP, 2015 - Line 14: "impact" should be
replaced by "implication", since the authors did not quantify the water cycle or the climate in
their experiments. - The short summary is very good indeed! (although I would recommend
changing the statement to 300 kDa, rather than 100 kDa.)
Author's response: We thank the referee for this comment. For consistency reasons with our
former studies, we prefer to keep upper case Nm.
As suggested by the reviewer, we changed "impact" to "implication".
References:

Després, V. R., Huffman, J. A., Burrows, S. M., Hoose, C., Safatov, A. S., Buryak, G., Fröhlich-
Nowoisky, J., Elbert, W., Andreae, M. O., Pöschl, U., and Jaenicke, R.: Primary biological
aerosol particles in the atmosphere: a review, Tellus B: Chemical and Physical Meteorology,
64, 15 598, 2012.
Failor, K. C., Schmale, D. G., Vinatzer, B. A., and Monteil, C. L.: Ice nucleation active bacteria
in precipitation are genetically diverse and nucleate ice by employing different mechanisms,
The ISME Journal, 11, 2740–2753, 2017.
Fröhlich-Nowoisky, J., Hill, T. C. J., Pummer, B. G., Yordanova, P., Franc, G. D., and Pöschl,
U.: Ice nucleation activity in the widespread soil fungus *Mortierella alpina*, Biogeosciences,
12, 1057–1071, 2015.
Hasegawa, Y., Ishihara, Y., and Tokuyama, T.: Characteristics of ice-nucleation activity in
*Fusarium avenaceum* IFO 7158, Bioscience, Biotechnology, and Biochemistry, 58, 2273–
2274, 1994.
Hoose, C. and Möhler, O.: Heterogeneous ice nucleation on atmospheric aerosols: a review of
results from laboratory experiments, Atmospheric Chemistry and Physics, 12, 9817–9854,
2012.
Humphreys, T. L., Castrillo, L. a., and Lee, M. R.: Sensitivity of partially purified ice nucleation
activity of *Fusarium acuminatum* SRSF616, Current Microbiology, 42, 330–338, 2001.
Kunert, A. T., Lamneck, M., Helleis, F., Pöhlker, M. L., Pöschl, U., and Fröhlich-Nowoisky,
J.: Twin-plate ice nucleation assay (TINA) with infrared detection for high-throughput droplet
freezing experiments with biological ice nuclei in laboratory and field samples, Atmospheric
Measurement Techniques, 11, 6327–6337, 2018.
Leslie, J. F. and Summerell, B. A.: The Fusarium Laboratory Manual, 2006.

Pouleur, S., Richard, C., Martin, J.-G., and Antoun, H.: Ice nucleation activity in *Fusarium*
*acuminatum* and *Fusarium avenaceum*, Applied and Environmental Microbiology, 1992.
Pummer, B. G., Atanasova, L., Bauer, H., Bernardi, J., Druzhinina, I. S., Fröhlich-Nowoisky,
J., and Grothe, H.: Spores of many common airborne fungi reveal no ice nucleation activity in
oil immersion freezing experiments, Biogeosciences, 10, 8083–8091, 2013.
Richard, C., Martin, J. G., and Pouleur, S.: Ice nucleation activity identified in some
phytopathogenic *Fusarium* species, Phytoprotection, 77,83–92, 1996.
Stopelli, E., Conen, F., Zimmermann, L., Alewell, C., and Morris, C. E.: Freezing nucleation
apparatus puts new slant on study of biological ice nucleators in precipitation, Atmospheric
Chemistry and Physics, 7, 129–134, 2014.
Tsumuki, H. and Konno, H.: Ice nuclei produced by *Fusarium* sp. isolated from the gut of the
rice stem borer, *Chilo suppressalis* Walker (Lepidoptera: Pyralidae), Bioscience,
Biotechnology, Biochemistry, 1994.
Tsumuki, H., Konno, H., Maeda, T., and Okamoto, Y.: An ice-nucleating active fungus isolated
from the gut of the rice stem borer, *Chilo suppressalis* Walker (Lepidoptera: Pyralidae), Journal
of Insect Physiology, 38, 119–125, 1992.
Tsumuki, H., Yanai, H., and Aoki, T.: Identification of ice-nucleating active fungus isolated
from the gut of the rice stem borer*, Chilo suppressalis* Walker (Lepidoptera: Pyralidae) and a
search for ice-nucleating active *Fusarium* species, Annals of the Phytopathological Society of
Japan, 61, 334–339, 1995.

---

## Author Comment (AC2) · 21 Oct 2019

**MS bg-2019-276, Kunert et al.: Highly active and stable fungal ice nuclei are widespread among Fusarium species**

We thank referee #2 for his/her constructive comments and suggestions, which are highly appreciated and have been taken into account upon revision of our manuscript. The comments and our answers are listed below (referee's comments marked with blue letters).

Specific comments:

Abstract:

Referee comment: Indicate the biological relevance of Fusarium and its ice nucleation activity. This is discussed well in the introduction but will help to bridge the first few sentences of the abstract.

Author's response: We thank the referee for this suggestion and included the following sentences in the abstract: "Ice nucleation activity in fungi was first discovered in the cosmopolitan genus *Fusarium*, which is widespread in soil and plants, has been found in atmospheric aerosol and cloud water samples, and can be regarded as the best studied IN-active fungus."

Moreover, we modified the following sentences: "The frequency and distribution of ice nucleation activity within *Fusarium*, however, remains elusive. Here, we tested more than 100 strains from 65 different *Fusarium* species for ice nucleation activity."

Methods 2.1:

Referee comment: How were the initial samples obtained? Could their original environment (crop vs. airborne, etc.) shed light on IN frequency?

Author's response: Samples from the USDA-ARS/Michigan State University were collected from crop tissue (sugar beet), and samples from the Schmale Laboratory at Virginia Tech were collected with unmanned aircraft systems. There is no detailed information available for the sources of the strains for the Kansas State University Teaching collection. We found IN activity in isolates from crop and air samples. For the air samples we cannot draw any conclusions from their original environment. A controlled comparison of IN frequency from samples collected in the air versus crop plants (and maybe even different types of crop plants) would be important, now that more IN-active species are known.

However, we added the following paragraph to section 2.1: "The strains from the USDA-ARS/Michigan State University were collected from crop tissue (sugar beet). All isolates were from field-grown beets and were obtained by hyphal tip transfer. The strains from the Schmale Laboratory at Virginia Tech were collected with unmanned aircraft systems (UASs or drones) equipped with remotely-operated sampling devices containing a *Fusarium* selective medium (e.g., Lin et al., 2013, 2014). All of the Schmale Laboratory strains were collected 100 m above ground level at the Kentland Farm in Blacksburg, Virginia, USA. Detailed information is not available for the sources of the strains for the Kansas State University Teaching collection. However, some of these strains are holotype strains referenced in Leslie and Summerell (2006)."

We extended Table S1 and provided additional information about sampling location and date.

Referee comment: Line 21: Additional, more recent, studies have contributed to this understanding of IN as well. (Failor et. al. 2017, Hanlon et al. 2017, Stopelli et al. 2017, 2015, Joly et al. 2014).

Author's response: We thank the referee for this remark and added the references to our manuscript.

Referee comment: Line 24-6: Failor et al. (2017) further expanded on known gammaproteobacteria IN.

Author's response: We changed the sentences as follows: "The best characterized biological IN are common plant-associated bacteria of the genera *Pseudomonas*, *Pantoea*, and *Xanthomonas* (Garnham et al., 2011; Govindarajan and Lindow, 1988; Graether and Jia, 2001; Green and Warren, 1985; Hill et al., 2014; Kim et al., 1987; Ling et al., 2018; Schmid et al., 1997; Wolber et al., 1986), and recently, an ice nucleation-active (IN-active) *Lysinibacillus* was found (Failor et al., 2017). The first identified IN-active fungi were strains of the genus *Fusarium* (Hasegawa et al., 1994, Pouleur et al., 1992, Richard et al., 1996, Tsumuki et al., 1992)."

Referee comment: Line 118: Was the range of incubation times necessary to reach a specified optical density? If so, that indication would be useful. If not, elaborate of reasoning for the times.

Author's response: Here, we did not mean that we tested these different incubation times. The sentence was meant to indicate the procedure considering all of the different replications that we used. For clarification, we changed "incubated" to "equilibrated".

Referee comment: Line 119: Be specific for the 0.5°C freezing point depression. Is it 0.5°C or 0.5±x °C.

Author's response: We added the calculations to the supplementary information.

We modified the sentence: "Note, that the aqueous extracts were prepared in 0.9 % NaCl solution, which could reduce the freezing temperatures by 0.5 °C based on theoretical calculations."

Results 3.1:

Referee comment: This would be an interesting point to note the original sampling locations for the various strains and could further demonstrate the cosmopolitan nature of these IN-active species should any tends be identified.

Author's response: We thank the referee for this comment, but as described before, we had only a few different sampling locations for both, the USDA-ARS/Michigan State University and samples from the Schmale Laboratory at Virginia Tech. For samples from the Kansas State University, we cannot specify the original sampling locations further as we obtained these samples from a culture collection.

Referee comment: Lines 154-5: This is a risky assumption to make. Prior to the Failor et al. study, all bacterial IN were thought to be proteinaceous. Exposing a selection of the species to high heat could support this claim.

Author's response: As many earlier studies already performed experiments with heat treatment of *Fusarium* IN, we initially refrained from repeating these experiments. The studies of Hasegawa et al. (1994), Pouleur et al. (1992), and Tsumuki and Konno (1994) only investigated some species of the genus *Fusarium*, and we agree with the referee that it is risky to generalize these findings to the newly found IN-active *Fusarium* species. Based on the suggestion of referee #2 and #3, we performed additional heat treatment experiments with four different
*Fusarium* species: *F. acuminatum*, *F. armeniacum*, *F. avenaceum*, and *F. langsethiae*.
We added a new Figure 4, and renumbered the other figures.
We included the following sentence in the abstract: "Heat treatment at 40 °C to 98 °C, however,
strongly reduced the observed IN concentrations, confirming earlier hypotheses that the INM
in *Fusarium* largely consists of a proteinaceous compound."
We modified the following sentence in the introduction: "Furthermore, the stability of
*Fusarium* IN upon exposure to ozone and nitrogen dioxide, under high and low or quickly
changing temperatures, and after short- and long-term storage under various conditions was
investigated."
We modified the following sentence in section 2.1: "For quantitative analysis, exposure
experiments, heat treatments, freeze-thaw cycles, as well as short- and long-term storage tests
a selection of IN-active tested strains was grown on full-strength potato dextrose agar (VWR
International GmbH, Darmstadt, Germany) first at room temperature for four to six days and
then at 6 °C for about four weeks."
We included the following sentences in section 2.2: "For heat treatment experiments, aliquots
of aqueous extracts of *F. acuminatum* 3-68, *F. armeniacum* 20970, *F. avenaceum* 2-106, and
*F. langsethiae* 19084 were incubated at 40 °C, 70 °C, and 98 °C, respectively, for one hour.
The IN concentration was determined using TINA."
We changed the following sentences in section 3.3: "They can be exposed to chemically
modifying agents like ozone and nitrogen dioxide, and physical stressors like high and low or
quickly changing temperatures. To investigate the stability of *Fusarium* IN, we performed
exposure experiments, heat treatments, freeze-thaw cycles, and long-term storage tests."
We included a new paragraph in section 3.3: "The stability of the INM in *Fusarium* was
investigated in heat treatment experiments. The ice nucleation activity was reduced
significantly at a 40 °C treatment (Fig. 4). Between 40 % and 90 % of IN were lost at this
temperature depending on the species, which supports the hypothesis that the INM in *Fusarium*
consists of a proteinaceous compound. A heat treatment at 70 °C reduced the ice nucleation
activity to less than 0.01 % compared to the initial level. Moreover, the initial freezing
temperature was shifted to lower temperatures indicating a breakdown of the large protein
aggregates. After a 98 °C treatment, we still found ice nucleation activity for all investigated
species except for *F.avenaceum* 2-106. The results are in agreement with previous studies,
which also reported a reduction in ice nucleation activity with increasing temperature in heat
treatment experiments (Hasegawa et al., 1994; Pouleur et al., 1992; Tsumuki and Konno, 1994).
The remaining activity after the 98 °C treatment, however, could indicate that post-translational
modifications like glycosylation and therefore polysaccharides could play a role in the ice
nucleation activity of *Fusarium*. Further systematic and chemical analysis studies are needed
for elucidation."
We included the following sentences in the conclusion: "A heat treatment of 40 °C reduced the
IN concentration significantly, supporting the hypothesis that the INM in *Fusarium* largely
consists of a proteinaceous compound. An involvement of polysaccharides, however, cannot
be excluded."

Referee comment: Lines 184-6: With the drastic decrease in activity after the 300,000 MWCO
filter and then again after 100,000, could the protein not be larger, but when damaged or broken
still retains some ice nucleation activity?
Author's response: If the INM in *Fusarium* is a single large protein, which breaks into small
parts upon filtration, we would expect based on Govindarajan and Lindow (1988) and Pummer
et al. (2015) a much lower initial freezing temperature of the filtrate than the temperature, which
we obtained in our experiments. The only small shift in the initial freezing temperature after
filtration suggests that small IN reassemble again to larger aggregates with similar activity than
before filtration. It is unlikely that a damaged or broken IN protein would show a similar activity
even if the broken parts would aggregate.
Referee comment: Lines 195-6: Why would single proteins in the atmosphere be unlikely?
Please elaborate on this statement.
Author's response: As hypothesized in Lines 184-185, the proteins tend to agglomerate, which
make it unlikely that individual proteins will enter the atmosphere. However; if an individual
protein would enter the atmosphere it would be in the nucleation mode size range of ∼ 6 nm.
These particles tend to grow by condensation of gaseous compounds (e.g., semi volatile organic
compounds, sulfates, water) and grow to particles in the Aitken mode size range. In this size
range further condensation and coagulation takes place and larger agglomerates are formed.
We included the following sentence to our manuscript: "Individual proteins with a diameter of
∼ 6 nm which may enter the atmosphere would be in the nucleation mode size range, where
particles tend to uptake gaseous compounds and grow to Aitken mode particles, which
themselves tend to coagulate to larger agglomerates (Seinfeld and Pandis, 1998)."
Referee comment: Line 216: Change ". . .and the fungus could safe energy." to ". . .and the
fungus could save energy.".
Author's response: Changed as suggested.
Referee comment: Figure 1. Inclusion of the positive control SnoMax curve would be beneficial
here. Any incidence of spontaneous freezing of the negative control should also be noted (if
any occurred with the methods you used).
Author's response: As the focus of this study is on fungal IN of *Fusarium*, we did not use
Snomax in any of the TINA experiments. The *Fusarium* strains themselves served as positive
controls based on the results of the initial screening (Table S1). Moreover, the correct
functionality of TINA including a Snomax curve is presented in Kunert et al. (2018).
For freezing tests, however, a negative control is essential. We added the information about the
negative controls and included the following sentences in the manuscript:
For the thermal cycler: "Aliquots of uninoculated DPY broth were used as negative controls,
which did not freeze in the investigated temperature interval."
For LINDA experiments: "As a negative control, a 0.9 % NaCl solution was added to three
uninoculated agar plates, and the freezing started below -14 °C."
For TINA experiments: "Pure water samples (0.1 μm filtered) served as a negative control for
each experiment. These did not freeze in the observed temperature interval."

*Pseudomonas syringae* CC94 was used as positive control for the initial screening using
LINDA as droplet freezing assay. We included the following sentence in section 2.3: "The
freezing temperatures ranged from -3.46 °C to -4.58 °C."
Referee comment: Figure 3. You note in the text that SnoMax has been shown to decrease after
exposure. Did you see this same result, or did you not use SnoMax because of this interaction?
Author's response: We showed in a previous study that the IN activity of Snomax decreased
after exposure to $O_3$ and $NO_2$ (Kunert et al. 2018). As this manuscript is focused on the IN
activity of *Fusarium*, we refrained from repeating the experiments.
References:

Govindarajan, A. G. and Lindow, S. E.: Size of bacterial ice-nucleation sites measured *in situ*
by radiation inactivation analysis, Proceedings of the National Academy of Sciences of the
United States of America, 85, 1334–1338, 1988.
Hasegawa, Y., Ishihara, Y., and Tokuyama, T.: Characteristics of ice-nucleation activity in
*Fusarium avenaceum* IFO 7158, Bioscience, Biotechnology, and Biochemistry, 58, 2273–
2274, 1994.
Kunert, A. T., Lamneck, M., Helleis, F., Pöhlker, M. L., Pöschl, U., and Fröhlich-Nowoisky,
J.: Twin-plate ice nucleation assay (TINA) with infrared detection for high-throughput droplet
freezing experiments with biological ice nuclei in laboratory and field samples, Atmospheric
Measurement Techniques, 11, 6327–6337, 2018.
Pouleur, S., Richard, C., Martin, J.-G., and Antoun, H.: Ice nucleation activity in *Fusarium*
*acuminatum* and *Fusarium avenaceum*, Applied and Environmental Microbiology, 1992.
Pummer, B. G., Budke, C., Niedermeier, D., Felgitsch, L., Kampf, C. J., Huber, R. G., Liedl,
K. R., Loerting, T., Moschen, T., Schauperl, M., Tollinger, M., Morris, C. E., Wex, H., Grothe,
H., Pöschl, U., Koop, T., and Fröhlich-Nowoisky, J.: Ice nucleation by water-soluble
macromolecules, Atmospheric Chemistry and Physics, 15, 4077–4091, 2015.
Stopelli, E., Conen, F., Zimmermann, L., Alewell, C., and Morris, C. E.: Freezing nucleation
apparatus puts new slant on study of biological ice nucleators in precipitation, Atmospheric
Chemistry and Physics, 7, 129–134, 2014.
Tsumuki, H. and Konno, H.: Ice nuclei produced by *Fusarium* sp. isolated from the gut of the
rice stem borer, *Chilo suppressalis* Walker (Lepidoptera: Pyralidae), Bioscience,
Biotechnology, Biochemistry, 1994.

---

## Author Comment (AC3) · 21 Oct 2019

**MS bg-2019-276, Kunert et al.: Highly active and stable fungal ice nuclei are widespread among Fusarium species**

We thank referee #3 for the review and positive assessment of our manuscript, and we are grateful for the detailed comments, which are very helpful for improving the manuscript. The comments and our answers are listed below (referee's comments marked with blue letters).

Referee comment: A main finding of this study is that filtration experiments suggest that the single cell-free Fusarium is smaller than 100 kDa. This is indeed very interesting and I wonder that the authors do not use the nomenclature of their own paper (Pummer et al., 2015) i.e. ice nucleating macromolecules (INM).
Author's response: We thank the referee for this remark and changed the nomenclature accordingly.

Referee comment: Indeed, water-soluble INMs have also been observed on many other primary biological aerosol particles (PABP) such as leaves, bark, pollen (Felgitsch et al., 2018), algae (Tesson et al., 2018), and bacteria (Failor et al., 2017). The sizes of these INM should be compared among each other, e.g. in a table.
Author's response: A precise comparison of the IN sizes in a table is rather difficult as most studies performed only a 0.2 µm filtration. A conclusion, which can be drawn upon these findings, is, that the IN are cell-free and stay active in solution.

We included the following sentence: "Moreover, biological INMs smaller than 200 nm were also found in various organisms e.g., other fungi (Fröhlich-Nowoisky et al., 2015; Pummer et al., 2015), leaves, bark, and pollen from birch trees (*Betula* spp.) (Felgitsch et al.,2018; Pummer et al., 2012), leaf litter (Schnell and Vali, 1973), some microalgae (Tesson and Šantl-Temkiv, 2018), strains of *Lysinibacillus* (Failor et al., 2017), and biological particles in the sea surface microlayer (Irish et al., 2019; Wilson et al., 2015)."

Referee comment: The same is true for the chemical composition and for the stability against oxidation. Also for other PBAPs, proteins and polysaccharides have been found as main components of INM and their stability is extraordinary as well. I also wonder if the authors have carried out heating experiments in order to destroy the ice nucleation activity of the proteins. Eventually, the heating was not successful due to the stability of INMs which would be important information since many colleagues use heating experiments to prove or unprove the presence of PBAP- INPs.
Author's response: Many earlier studies already performed heat treatment experiments on different IN-active *Fusarium* species and strains, including strains of *F. acuminatum* and *F. avenaceum*, consistently showing a small reduction of ice nucleation activity after heating to 40 °C and a bigger loss after heating to 70 °C (Hasegawa et al. (1994), Pouleur et al. (1992), and Tsumuki and Konno (1994)). Thus, we expected similar results from strains and species of the genus *Fusarium* and we initially refrained from repeating these experiments. Based on the suggestion of referee #2 and #3, we performed additional heat treatment experiments with four different *Fusarium* species: *F. acuminatum*, *F. armeniacum*, *F. avenaceum*, and *F. langsethiae*.

We added a new Figure 4, and renumbered the other figures.

We included the following sentence in the abstract: "Heat treatment at 40 °C to 98 °C, however, strongly reduced the observed IN concentrations, confirming earlier hypotheses that the INM in *Fusarium* largely consists of a proteinaceous compound."

We modified the following sentence in the introduction: "Furthermore, the stability of *Fusarium* IN upon exposure to ozone and nitrogen dioxide, under high and low or quickly changing temperatures, and after short- and long-term storage under various conditions was investigated."

We modified the following sentence in section 2.1: "For quantitative analysis, exposure experiments, heat treatments, freeze-thaw cycles, as well as short- and long-term storage tests a selection of IN-active tested strains was grown on full-strength potato dextrose agar (VWR International GmbH, Darmstadt, Germany) first at room temperature for four to six days and then at 6 °C for about four weeks."

We included the following sentences in section 2.2: "For heat treatment experiments, aliquots of aqueous extracts of *F. acuminatum* 3-68, *F. armeniacum* 20970, *F. avenaceum* 2-106, and *F. langsethiae* 19084 were incubated at 40 °C, 70 °C, and 98 °C, respectively, for one hour. The IN concentration was determined using TINA."

We changed the following sentences in section 3.3: "They can be exposed to chemically modifying agents like ozone and nitrogen dioxide, and physical stressors like high and low or quickly changing temperatures. To investigate the stability of *Fusarium* IN, we performed exposure experiments, heat treatments, freeze-thaw cycles, and long-term storage tests."

We included a new paragraph in section 3.3: "The stability of the INM in *Fusarium* was investigated in heat treatment experiments. The ice nucleation activity was reduced significantly at a 40 °C treatment (Fig. 4). Between 40 % and 90 % of IN were lost at this temperature depending on the species, which supports the hypothesis that the INM in *Fusarium* consists of a proteinaceous compound. A heat treatment at 70 °C reduced the ice nucleation activity to less than 0.01 % compared to the initial level. Moreover, the initial freezing temperature was shifted to lower temperatures indicating a breakdown of the large protein aggregates. After a 98 °C treatment, we still found ice nucleation activity for all investigated species except for *F.avenaceum* 2-106. The results are in agreement with previous studies, which also reported a reduction in ice nucleation activity with increasing temperature in heat treatment experiments (Hasegawa et al., 1994; Pouleur et al., 1992; Tsumuki and Konno, 1994). The remaining activity after the 98 °C treatment, however, could indicate that post-translational modifications like glycosylation and therefore polysaccharides could play a role in the ice nucleation activity of *Fusarium*. Further systematic and chemical analysis studies are needed for elucidation."

We included the following sentences in the conclusion: "A heat treatment of 40 °C reduced the IN concentration significantly, supporting the hypothesis that the INM in *Fusarium* largely consists of a proteinaceous compound. An involvement of polysaccharides, however, cannot be excluded."

Comment:

Referee comment: The abbreviation "IN" has been used in a confusing way. In the text it means "ice nuclei" but also means "ice nucleation" and "ice nucleating". I recommend using "INP" for "ice nucleating particles" and write the full words in all other cases.

Author's response: We thank the referee for this comment. We clearly defined ice nuclei as IN and ice nucleation-active as IN-active in the abstract and the introduction. We used the abbreviation IN for ice nuclei in our former studies (Després et al., 2012, Fröhlich-Nowoisky et al., 2015, 2016, Kunert et al., 2018, Pummer et al., 2015), and for consistency reasons we

prefer to keep it this way. To avoid misunderstanding, we changed "IN activity" to "ice nucleation activity".

References:

Després, V. R., Huffman, J. A., Burrows, S. M., Hoose, C., Safatov, A. S., Buryak, G., Fröhlich-Nowoisky, J., Elbert, W., Andreae, M. O., Pöschl, U., and Jaenicke, R.: Primary biological aerosol particles in the atmosphere: a review, Tellus B: Chemical and Physical Meteorology, 64, 15 598, 2012.

Fröhlich-Nowoisky, J., Hill, T. C. J., Pummer, B. G., Yordanova, P., Franc, G. D., and Pöschl, U.: Ice nucleation activity in the widespread soil fungus *Mortierella alpina*, Biogeosciences, 12, 1057–1071, 2015.

Fröhlich-Nowoisky, J., Kampf, C. J., Weber, B., Huffman, J. A., Pöhlker, C., Andreae, M. O., Lang-Yona, N., Burrows, S. M., Gunthe, S. S., Elbert, W., Su, H., Hoor, P., Thines, E., Hoffmann, T., Després, V. R., and Pöschl, U.: Bioaerosols in the Earth system: Climate, health, and ecosystem interactions, Atmospheric Research, 182, 346–376, 2016.

Hasegawa, Y., Ishihara, Y., and Tokuyama, T.: Characteristics of ice-nucleation activity in *Fusarium avenaceum* IFO 7158, Bioscience, Biotechnology, and Biochemistry, 58, 2273–2274, 1994.

Kunert, A. T., Lamneck, M., Helleis, F., Pöhlker, M. L., Pöschl, U., and Fröhlich-Nowoisky, J.: Twin-plate ice nucleation assay (TINA) with infrared detection for high-throughput droplet freezing experiments with biological ice nuclei in laboratory and field samples, Atmospheric Measurement Techniques, 11, 6327–6337, 2018.

Pouleur, S., Richard, C., Martin, J.-G., and Antoun, H.: Ice nucleation activity in *Fusarium acuminatum* and *Fusarium avenaceum*, Applied and Environmental Microbiology, 1992.

Pummer, B. G., Budke, C., Niedermeier, D., Felgitsch, L., Kampf, C. J., Huber, R. G., Liedl, K. R., Loerting, T., Moschen, T., Schauperl, M., Tollinger, M., Morris, C. E., Wex, H., Grothe, H., Pöschl, U., Koop, T., and Fröhlich-Nowoisky, J.: Ice nucleation by water-soluble macromolecules, Atmospheric Chemistry and Physics, 15, 4077–4091, 2015.

Tsumuki, H. and Konno, H.: Ice nuclei produced by *Fusarium* sp. isolated from the gut of the rice stem borer, *Chilo suppressalis* Walker (Lepidoptera: Pyralidae), Bioscience, Biotechnology, Biochemistry, 1994.